



# Real-time NMR spectroscopy in the study of biomolecular kinetics and dynamics

György Pintér[1], Katharina F. Hohmann[1], J. Tassilo Grün[1], Julia Wirmer-Bartoschek[1], Clemens Glaubitz[2], Boris Fürtig[1], Harald Schwalbe[1]

[1]Institute for Organic Chemistry and Chemical Biology, Center for Biomolecular Magnetic Resonance (BMRZ), Johann Wolfgang Goethe-Universität Frankfurt, Frankfurt, 60438, Germany
[2]Institute for Biophysical Chemistry, Center for Biomolecular Magnetic Resonance (BMRZ), Johann Wolfgang Goethe-Universität Frankfurt, Frankfurt, 60438, Germany

*Correspondence to*: Harald Schwalbe (schwalbe@nmr.uni-frankfurt.de)

**Abstract.** The review describes the application of NMR spectroscopy to study kinetics of folding, refolding and aggregation of proteins, RNA and DNA. Time-resolved NMR experiments can be conducted in a reversible or an irreversible manner. In particular irreversible folding experiments pose large requirements on (i) the signal-to-noise due to the time limitations and (ii) on synchronizing the refolding steps. Thus, this contribution discusses the application of methods for signal-to-noise

increases including dynamic nuclear polarization, hyperpolarization and photo-CIDNP for the study of time-resolved NMR studies. Further, methods are reviewed ranging from pressure- and temperature-jump, light induction and rapid mixing to induce rapidly non-equilibrium conditions required to initiate folding.

## 1 Introduction

In 1993, the journal Current Opinion in Structural Biology published a special edition on protein-nucleic acid interactions

edited by D. Moras and S. Philipps and on protein folding edited by C. Dobson. The edition featured an editorial article by Phillips and Moras (1993) on protein-nucleic acid interactions, and reviews on transcription factor structure and DNA binding by Wolberger (1993), zinc-finger proteins by Berg (1993), DNA repair enzymes by Morikawa (1993), restriction endonucleases and modification methylases by Anderson (1993), DNA- and RNA-dependent DNA polymerases by Steitz (1993), aminoacyl-tRNA synthetases by Cusack (1993), work on ribosomes by Yonath and Franceschi (1993) and

contributions by Robert (Rob) Kaptein on "protein-nucleic acid interaction by NMR" (Kaptein, 1993). This first part on protein-nucleic acids complexes was accompanied by a second part, introduced in the editorial article by Chris Dobson (1993), on protein folding with contributions by Dyson and Wright (1993) on peptide conformation and protein folding, on denatured states of proteins by Shortle (1993), on principles of protein stability by Fersht and Serrano (1993), H/D exchange experiments by Baldwin (1993), molecular simulation of peptide and protein folding by Brooks (1993), on protein folding by

Dill (1993), accessory protein in protein folding by Jaenicke (1993), and antibody-antigen interaction by Wilson and Stanfield (1993).



Being invited to contribute to this edition of Current Opinion in Structural Biology nicely documents the eminent role that Rob Kaptein played as NMR spectroscopist in Structural Biology early on, in the heroic age of biomolecular NMR spectroscopy. Beyond the NMR community, his work was highly influential in the broad field of Structural Biology and

known in this broad community. His main research focus, pursued together with Rolf Boelens in Groningen and in Utrecht, is the development of NMR spectroscopy for the determination of structure and dynamics of biomacromolecules, in particular for protein-DNA-complexes. He was among the first to solve the structure of a sizeable protein, and he was among the first to push NMR towards 3D spectroscopy. Rob Kaptein is thus an NMR pioneer in bio-NMR. In addition to pushing the capabilities of NMR structure determination, the use of Photo-CIDNP is intimately linked to Rob Kaptein.

The fundamental discovery of CIDNP goes back to J. Bargon and H. Fisher (Bargon et al., 1967) (see (Bargon, 2006) citations therein) and, independently, by Ward and Lawler (1967). P. Hore provided key contributions (Hore and Broadhurst, 1993), but it was the work of Kaptein (Berliner and Kaptein, 1980; Buck et al., 1980; Kemmink et al., 1986a; Redfield et al., 1985; Scheek et al., 1979) that brought about the application of Photo-CIDNP to biomolecular NMR spectroscopy. The explanation via the radical-pair mechanism and its description via the Kaptein rules are one prime example for his seminal

contributions in this field (see (Kaptein, 1975)). They were independently confirmed by Closs and Closs (1969a, 1969b). The CIDNP studies on flavins (Kaptein and Oosterhoff, 1969) initiated the application of Photo-CIDNP to proteins (Kaptein et al., 1978) and brought about its broad use in biomolecular NMR spectroscopy (Hore and Kaptein, 1982) with numerous examples (Berliner and Kaptein, 1980; Buck et al., 1980; Kemmink et al., 1986a; Redfield et al., 1985; Scheek et al., 1979). Kaptein showed the general applicability of Photo-CIDNP NMR in probing the accessibility of aromatic amino acids in

proteins to fluorescent dyes and the concurrent manifold enhancement of NMR signal intensity through photo-induced dynamic nuclear polarization. To conduct these Photo-CIDNP experiments, Kaptein coupled high power laser irradiation within the NMR tube in the NMR magnet and integrated light illumination into the NMR experiments. In situ-illumination of fluorescent dyes leads to signal enhancement of the aromatic acceptor amino acids tryptophan, in particular, tyrosine, phenylalanine and histidine, but also nucleobases in RNA and DNA. The possibility to couple laser light into the NMR tube

and excite fluorophores homogeneously in samples dissolved in NMR tubes paved the way to utilize endogenous chromophors in proteins that are rigidly attached to a protein, as trigger for light-induced changes of protein conformation. In Kaptein's lab, one culmination of such approach was the study of photo-active yellow protein, which yielded important information on light-activated states of proteins, not obtainable by other structural biology techniques.

The research of Kaptein and seminal work pioneered in his group provided prerequisites for research interests in the group of

the authors in developing and applying time-resolved NMR to a number of different systems including protein, RNA and DNA folding, refolding and aggregation. This contribution will thus focus on this topic. Anecdotally, a number of things should be added here: The work in our group was greatly influenced by Rob Kaptein, but also during a postdoctoral stay of one of us (hs) in Oxford by Peter Hore. Peter Hore conducted his postdoc with Kaptein, and made great contributions, and carried on the torch of photo-CIDNP at times, where the research focus of Kaptein and Boelens shifted more into

biomolecular NMR spectroscopy. Here, he shifted his focus to the development of protein structure determination (Kaptein



et al., 1985) in parallel with K. Wüthrich and 3D NMR spectroscopy (Vuister et al., 1990) in parallel with C. Griesinger and R.R. Ernst. Heinz Rüterjans, who for long hold the position for NMR spectroscopy in Biochemistry at Goethe-University Frankfurt, joined Kaptein's lab in Groningen, then still in Münster, to use Photo-CIDNP to study the interaction of the Lac headpiece and DNA, which became a new research with excellent possibilities in the Netherland. Jacques van Boom had just developed DNA synthesis in the Organic Chemistry department of Leiden University, where Rob had done his PhD thesis. Kaptein formed a team with the biochemist Ruud Scheek. Scheek established headpiece and DNA purification and carried out the first NMR experiments of DNA complexes and worked on DNA assignment. Erik Zuiderweg from the Hilbers group in Nijmegen was in the team. He assigned the NMR spectra of the Lac headpiece and restrained molecular dynamics simulation (at the time!). And, of course, Rolf Boelens, who pushed the limits of two-dimensional NMR on complexes of Lac and DNA. These studies defined the size limitation of 2D NMR at the time. Heinz Rüterjans' project was thus intimately linked to the projects that formed the basis of success of the group in Utrecht around Rob Kaptein and Rolf Boelens. It should also be mentioned that one of us (hs) conducted his first laser-induced folding reactions on alpha-lactalbumin together with Till Kühn in Utrecht using the laser installations at the Utrecht European NMR Large Scale facility in 1999. This review will thus summarize approaches to time-resolved NMR spectroscopy and coupling of methods to increase signal-to-noise in NMR for such time-resolved experiments.

## 2 Techniques to trigger real-time NMR experiments

NMR spectroscopy is unique in studying the kinetics of reactions and conformational transitions, including biomolecular folding and refolding with atomic site resolution, and ever since the dawn of NMR, such applications have been pursued. Biomolecular folding can be studied at equilibrium or under non-equilibrium conditions and the theory describing the peculiar appearance of Fourier-transformed NMR spectra recorded during fast irreversible non-equilibrium reactions has been developed early on (Kühne et al., 1979). While equilibrium studies focus on characterization of conformational transitions in the microsecond-to-millisecond timescale involving NOESY-type experiments (Evans et al., 1989), lineshape analysis (Evans et al., 1989; Huang and Oas, 1995) or relaxation dispersion (Korzhnev et al., 2004), non-equilibrium studies focus on slower biomolecular folding transitions. The induction of non-equilibrium conditions can be conducted in an irreversible manner. A prime example for the kinetic studies under irreversible conditions are experiments that utilize a rapid change in solution conditions, a so-called mixing step, and subsequent spectroscopic quantification of the build-up of a new equilibrium under the conditions after mixing. This is commonly accomplished by mixing-based NMR approaches comprising stopped-flow (Frieden et al., 1993; McGee and Parkhurst, 1990), rapid injection (Balbach et al., 1995) as well as mixing probe technologies ((Spraul et al., 1997) and reviewed in (Schlepckow et al., 2011)) or laser-based experiments (Kühn and Schwalbe, 2000). The time resolution of the experiments is mainly determined by acquisition of sample homogeneity. Dead-times can be as short as 50 milliseconds nowadays (Mok et al., 2003a; Schlepckow et al., 2008). A further extension of time-resolved NMR is to utilize freeze-quenching as an approach that couples rapid induction of





biomolecular folding with rapid freezing of the acquired conformational ensemble at appropriate time points after folding
induction to study its properties by solid-state NMR spectroscopy (Etzkorn et al., 2007; Jeon et al., 2019).

A fundamentally different approach to studying biomolecular folding and refolding reactions is to change the state
parameters pressure or temperature to rapidly induce non-equilibrium conditions. If the pressure- or T-jumps do not induce
alterations in the macromolecular system of interest, in particular if temperature- or pressure-induced aggregation can be
circumvented, then changes of state parameters are reversible and can thus be applied multiple times, allowing for
sophisticated multidimensional NMR detection schemes. In the following chapter 2, we discuss the available methods to
conduct such time-resolved NMR experiments.

## 2.1 Rapid mixing

The first real-time NMR investigations of protein folding induced by rapid mixing have been conducted in 1988.
Applications focused on coupling of protein folding and hydrogen exchange (Elove et al., 1994; Radford et al., 1992; Roder
et al., 1988; Udgaonkar and Baldwin, 1988) and became particularly popular in the mid-90s (Balbach et al., 1996, 1995;
Hoeltzli and Frieden, 1995; Kiefhaber et al., 1995; Van Nuland et al., 1998). Rapid mixing experiments typically use a setup
with a Teflon transfer line, filled with buffer solution that connects the NMR tube with an injection piston outside the
magnet. The sample solution is loaded into the transfer line and separated from the solution in the NMR tube by an air
bubble that precludes premature mixing at a liquid-liquid interface (see Figure 1). A pneumatic trigger induces rapid
injection (Van Nuland et al., 1998). This simple rapid-mixing setup has been revised to an *in situ* device ready to use in
conventional NMR probes with a dead time after injection of 50 ms (Mok et al., 2003b). More recently, a 3D-printed rapid-
mixing device with optimized injection has been reported (Franco et al., 2017a). A removable injector allows using smaller
sample volumes and minimizing the disturbance of the magnetic field homogeneity. While originally not intended for
measuring kinetics, also dissolution DNP (see below) uses rapid mixing to inject polarized water into the NMR tube
allowing triggering kinetics by adding a folding cofactor simultaneously with the polarized water, as shown recently (Chen
et al., 2013; Novakovic et al., 2020b).



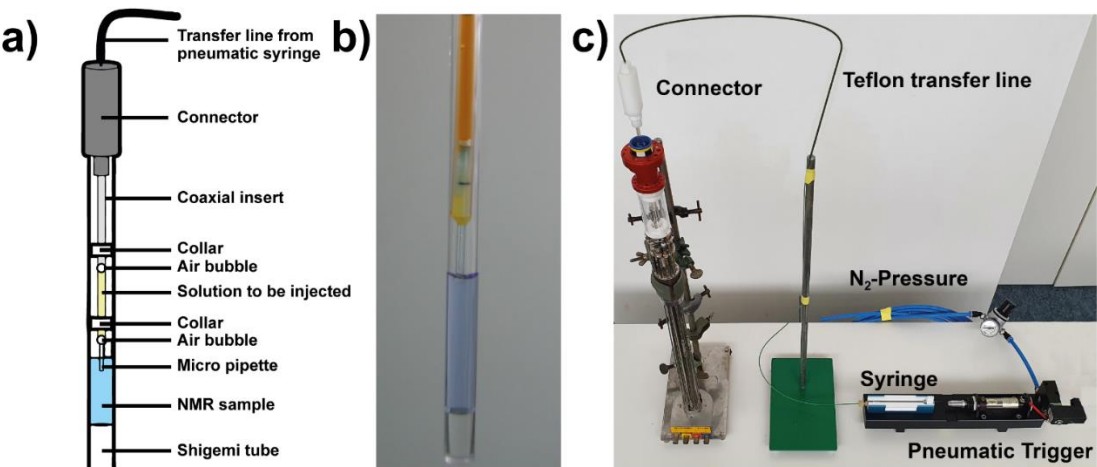

**Figure 1: Setup to trigger real-time NMR experiments with *in situ* rapid-mixing. a) Schematic representation of the rapid-mixing device introduced by Mok et al., 2003b. b) Shigemi tube with injection insert. Air bubbles prevent premature mixing via a liquid-liquid interface. c) Rapid-mixing setup used at BMRZ, Frankfurt. The sample is connected via a Teflon transfer line to an external syringe that induces rapid mixing after a pneumatic trigger.**

Biomacromolecules can be unfolded in many different ways (Fürtig et al., 2007a; Roder et al., 2004). Proteins can be chemically denatured using high concentrations (6-8 M) of guanidinium chloride (GdnCl) (Logan et al., 1994; Zeeb and Balbach, 2004) or urea (Egan et al., 1993; Neri et al., 1992; Schwalbe et al., 1997), but also organic solvents including TFE or DMSO (Buck, 1998; Buck et al., 1995, 1993; Nishimura et al., 2005). The (re-)folding of chemically denatured proteins can then be initiated with a rapid dilution into native buffer conditions (Balbach et al., 1995) or vice versa for unfolding of native proteins (Kiefhaber et al., 1995). Alternatively, a rapid pH-change can be used to re- or de-nature proteins (Balbach et al., 1996; Corazza et al., 2010; Dobson and Hore, 1998a; Schanda et al., 2007; Zeeb and Balbach, 2004). The rapid mixing design introduced by Mok *et al.* (Mok et al., 2003b) has also found widespread application for studies on folding of nucleic acids (see below).

## 2.2 Pressure jump

Pressure is a physical state parameter that can influence the conformation of biomolecules. High pressure can denature proteins. This process is usually reversible and upon release of pressure the protein folds back to its native state. The required pressure can be adjusted by use of chaotropic agents to lower the overall stability, or by introducing specific mutations to introduce internal cavities in the folded structure (Bouvignies et al., 2011; Mulder et al., 2001). Static high pressure NMR spectroscopy is a long established method to assess the thermodynamic profile of proteins (Balbach et al., 2019), and allows detailed thermodynamic characterization of the energy landscape (Akasaka et al., 2013; Roche et al., 2019). There are further reviews about equilibrium high-pressure measurements as reviewed in (Caro and Wand, 2018; Nguyen and Roche, 2017; Roche et al., 2017). Here, we focus on more recent developments regarding the rapid change of pressure inside the spectrometer to study kinetics of biomolecules, especially protein folding kinetics.



High pressure NMR measurements require a special NMR tube (made from either quartz, sapphire or zirconia) that can withstand high pressures up to a few thousand bar. The most often used ones are made from zirconium oxide, as they are commercially available and are specified up to a maximum 3000 bar (Daedalus Innovations LLC). The pressure is usually realized by use of an external hydrostatic pressure pump connected via pressure withstanding tubing to the sample inside the
spectrometer. Usually, mineral oils are used to transmit the pressure providing phase separation from the typical water samples. The up to now most advanced setup was developed by (Charlier et al., 2018a) and the schematic is show in Figure 2. The system uses a high pressure and an atmospheric pressure reservoir, connected through a hydraulic valve to the sample. By opening the valve from the high pressure reservoir, the pressure is rapidly equilibrated in the sample to the high pressure, while the pressure rapidly drops in the sample by changing to the low pressure reservoir. The oil reservoir is kept in a close
system under nitrogen atmosphere to avoid oxidation.

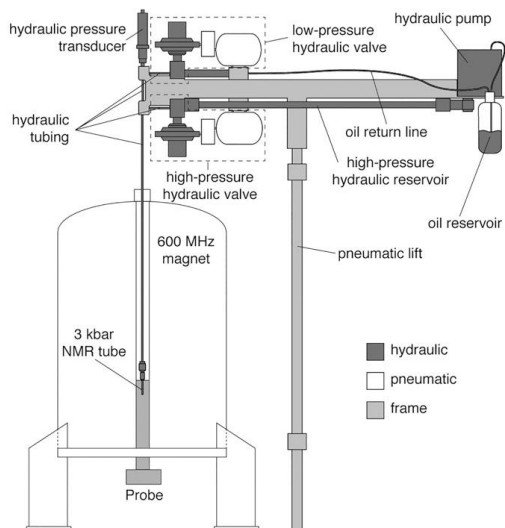

**Figure 2: Schematic representation of the rapid pressure-jump NMR apparatus, developed by (Charlier et al., 2018a). The apparatus is mounted onto a frame with pneumatic lift (light gray) to adjust the height to any spectrometer. The pressure apparatus is a closed system under N$_2$ atmosphere and uses a high-pressure and atmospheric pressure reservoir to increase or to**
**decrease the pressure, correspondingly inside the sample stored in a Zirconia tube inside the spectrometer. Reprinted with permission from (Charlier et al., 2018a).**

The major advantage of pressure jump compared to other methods to study protein folding and energy landscape lies in the reversibility of the induced conformational transition. In combination with the shortest time (1-5 ms) requirement to change between folding and unfolding conditions this method allows complex NMR experiment designs to study in details the
folding pathways, mechanism and even the structure of intermediates.

### 2.3 Light induction

Folding can be initiated photochemically by irradiation within the NMR spectrometer. Laser irradiation of biomolecules within the NMR-spectrometers has been introduced by Kaptein in photo-CIDNP NMR (Kaptein et al., 1978), before first





real-time folding applications have been conducted. In folding applications, high-power laser irradiation (up to 8-10 W
primary output) is coupled to the NMR-spectrometer by a quartz fibre ending in the NMR-tube within the spectrometer.
Figure 3 shows the setup of two lasers coupled to an NMR spectrometer as it is used at BMRZ in Frankfurt. To achieve
reasonable irradiation times (typically between 0.2-4s) depending on the folding rate to be observed, not only the applied
power is important but also the homogeneous light illumination within the sample. Different methods that advance the setup
presented by Berliner (Scheffler et al., 1985), such as using a cone shaped quartz tip (Kühn and Schwalbe, 2000) or stepwise
tapered (Kuprov and Hore, 2004) or sandblasted quartz fibre ends (Feldmeier et al., 2013) have been introduced to achieve
this.

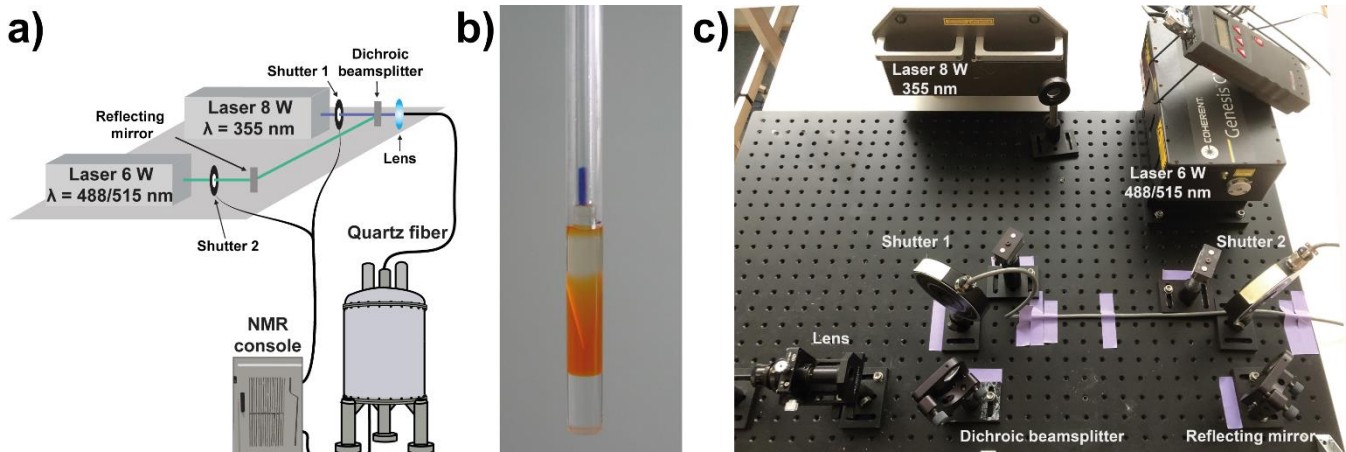

**Figure 3: Setup to trigger real-time NMR experiments with *in situ* light induction. a) Schematic representation of a two-source
setup for high-power laser irradiation connected to a NMR tube via a quartz fiber. The shutters can be directly triggered via the
NMR console/pulse programme. b) Shigemi tube with cone shaped quartz tip for homogeneous sample illumination (Kühn and
Schwalbe, 2000). The glass fiber can be inserted into the plunger. c) Two-wavelength laser setup installed at BMRZ, Frankfurt.**

Most importantly, the approach relies on the presence of a chromophore within the NMR sample. This can either be a
biomolecule carrying a photosensitive group such as the yellow protein (Derix et al., 2003), or folding can be initiated by
release of cofactors from photo-labile chelators (Kühn and Schwalbe, 2000), from caged ligands (Buck et al., 2007) or from
photo-labile precursors that cage biomolecular conformation (Wenter et al., 2005).

### 2.3 Temperature jump

Next to pressure, temperature is the second thermodynamic state parameter. It is coupled to the free enthalpy of a
conformational equilibrium. Thus, the change of temperature was one of the first methods to initiate changes in biomolecular
systems and ultrafast T-jump experiments have been introduced by M. Eigen and awarded with the Nobel prize in Chemistry
in 1967. The first application of temperature-jump in combination with NMR spectroscopy was the study of proline *cis-trans*
isomerization in oligopeptides as an alternative to the jump in pH(D) by Wüthrich (Grathwohl and Wüthrich, 1981).



Additionally, temperature changes can initiate refolding of biomolecules exhibiting temperature-dependent conformations (Reining et al., 2013; Rinnenthal et al., 2010), denature proteins at high temperature, or refold cold-denatured proteins.

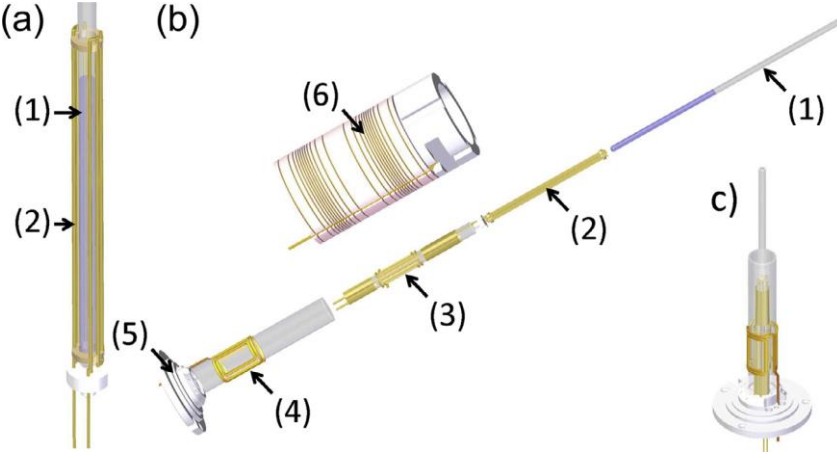

**Figure 4: Schematic view of the coil assemblies in the T-jump probehead developed by (Rinnenthal et al., 2015). (a) Cage coil/wire capacitor for rf-heating with sample tube (2.5 mm), (b) exploded view of the overall assembly with (1) sample tube, (2) 2.5 mm cage coil, (3) 5 mm double tuned ($^1$H and $^2$H) rf NMR saddle coil, (4) 10 mm ($^{15}$N) rf NMR saddle coil, (5) coil insert base and connection to main probe body, (6) z gradient coil. (c) Full assembly of coils in the T-jump probe (z gradient system not shown). Reproduced with permission from (Rinnenthal et al., 2015).**

Later, different technical setups were developed to speed up the temperature change to study dynamic changes with faster reaction rates. A list of the different techniques to achieve a temperature jump is given in Table 1. In this table, important parameters are described for each technique. Out of all the different T-jump techniques, microwave (MW) and radio-frequency (RF) heating proved to be the most suitable for biomolecular NMR. In both cases, inductive and dielectric heating effects take place, the latter is the major factor and couples well to lossy samples (salt-containing aqueous samples). RF heating allows easy coupling to the spectrometer, due to built-in RF generators and amplifier system. It can reach relatively fast heating with 20 K/sec, although slower than MW setups, but offers a more homogeneous heating profile which is required for high resolution NMR spectroscopy.

**Table 1 Comparison of different T-jump initiation methods used in combination with NMR spectroscopy.**

| T-jump method | Heating speed | Homogeneity | Temperature range | Special requirement | Temperature stability | BioNMR Application |
|---|---|---|---|---|---|---|
| Laser (Ernst et al., 1996; Ferguson et al., 1994) | Fast | Low | Excellent (up to few hundred K) | Laser setup | No | - |
| Gas-heating (Akasaka et al., 1990) | Slow (in both direction) | Good | Small | Vibration stabilization | Yes | Refolding of RNase A from thermal denatured state (Akasaka et |



| | | | | | | |
|---|---|---|---|---|---|---|
| | | | | | | al., 1991) |
| Flow system | | Moderate | Good | Moderate | Stop-flow system | Yes | Folding of heat denatured RNase A (Yamasaki et al., 2013) |
| Dielectric heating | Microwave (> 1 GHz) | Fast | Moderate (Good with mechanical mixing device) (Kawakami and Akasaka, 1998) | Moderate-Good | Special coil design | No | RNase A denaturation (Naito et al., 1990) |
| | Radio-frequency (< 1 GHz) | Moderate-fast | Good | Moderate | Special coil design | In combination with gas heating | Folding of cold denatured barstar (Pintér and Schwalbe, 2020; Rinnenthal et al., 2015) |

The latest RF heating setup, shown in Fig. 4, described in the literature (Rinnenthal et al., 2015) uses a built-in RF coil to initiate the jump with an additional optimized gas-heating to stabilize final temperature. The range of the temperature jump can be adjusted by the number of heating RF pulses applied, and for longer measurements, the gas-heating provides stability at the final temperature. The suitability of this setup to study folding mechanism of proteins has been demonstrated on the cold-denatured barstar, where the temperature jump initiated the complete reversible refolding of the protein.

**2.4 General pulse scheme for RT-NMR**

In the context of real-time NMR, a number of aspects within pulse sequences has to be conceptualized. Firstly, the timing of NMR excitation, synchronous triggering of folding, and the correlation of NMR coherences or polarizations have to be designed (shown in Fig. 5). Secondly, the best excitation pulses and detection schemes have to be applied. The pulse sequences used to measure time-resolved NMR experiments depend on the trigger and on the system under study. These can

be divided into two major groups: non-reversible (Fig. 5a) or reversible systems (Fig. 5b-c). In both cases, before initiating the kinetic experiment, reference spectra are recorded. For non-reversible systems, the basic scheme is a simple trigger after which a series of 1D-NMR spectra is recorded, allowing the best time resolution. While two-dimensional experiments can provide higher chemical shift resolution, they can only be utilized for slow kinetic measurements. Depending on the timescale of the observed kinetics, $^{15}N/^{13}C-^{1}H$ correlation spectra can be measured. Modifications to these experiments can

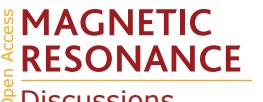

speed up the recording and further increase the time resolution. These techniques include different SOFAST and BEST-HMQC techniques (Favier and Brutscher, 2011; Schanda et al., 2006, 2005; Schanda and Brutscher, 2005) with longitudinal relaxation optimization (Farjon et al., 2009), Hadamard frequency encoding (Schanda and Brutscher, 2006) or ultrafast approaches (Gal et al., 2007) or in case of NOESY experiments Looped-PROjected SpectroscopY (L-PROSY) (Novakovic et al., 2020a, 2018). Furthermore, several of the pulse sequences can be combined with non-uniform sampling (NUS)
(Gołowicz et al., 2020) to further reduce the final measurement time up to a few seconds.

For reversible systems, two dimensional experiments can be recorded with the same time resolution as 1D-NMR experiments. To achieve this, only one increment of the indirect dimension is recorded in every kinetic experiment and time incrementation in the indirect dimension is achieved after each new folding trigger. Repeating the experiments and measuring the required indirect dimension points and finally concatenating together the corresponding time points, kinetic
measurements with high temporal and spectral resolution can be achieved (Alderson et al., 2017; Harper et al., 2004; Kremer et al., 2011; Naito et al., 1990).



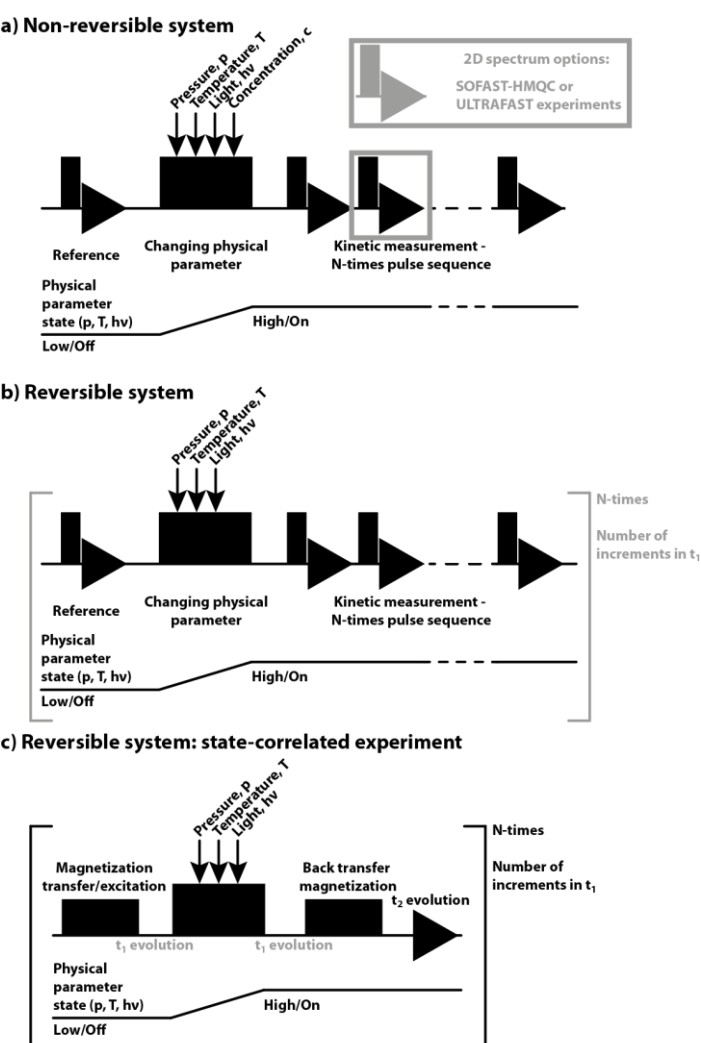

**Figure 5: General pulse schemes used for RT-NMR measurements. In all cases first reference spectrum is measured, followed by a rapid trigger using corresponding physical parameter change. In a) general scheme is shown for non-reversible systems and in b) reversible systems are shown. In c) SC-spectrum experiments are shown, suitable for reversible systems to study rapid changes (< 1 sec). $T_1$ evolution time can be introduced either before or after the trigger part.**

It is noteworthy that reversible systems in combination with magnetization transfer between starting state one (at the start) and state two (new equilibrium) allow recording of a so called state-correlated (SC) spectrum. In SC-spectrum, the pulse sequence starts with magnetization transfer to other nuclei before the actual physical parameter (light, pressure or temperature) is changed. The final detection takes place already in the new equilibrium state. The limiting factor for such application is the speed of the physical parameter change, as it has to be faster than the $T_1$ relaxation time. Additionally, double jump experiments can also be utilized to observe alternative folding pathways if rapid changes between state one and state two in both directions are possible (Charlier et al., 2018c; Kremer et al., 2011; Pintér and Schwalbe, 2020).



## 3 Signal enhancement

It is beyond the scope of this article to cover the physical principles of the plethora of signal-to-noise enhancement approaches in NMR spectroscopy. Time-resolved experiments, however, provide very stringent requirements on signal-to-noise as the kinetics of structural transitions in biomolecular folding define the time window that is available. In the following, we will discuss the coupling of time-resolved experiments to dynamic nuclear polarization (DNP) in solid-state NMR (Becerra et al., 1993; Corzilius, 2020), to hyperpolarization (Ardenkjær-Larsen et al., 2003; Ragavan et al., 2011) and

to photo-CIDNP experiments in liquid-state NMR.

### 3.1 DNP

Dynamic nuclear polarization describes a set of mechanisms by which nuclear non-Boltzmann magnetization is created (for a recent review see (Corzilius, 2020)). So far, it works best for solid-state NMR and especially polarisation transfer via the cross-effect using biradicals as polarisers has progressed from the proof-of-concept stage to numerous applications.

Enhancements of up to 150-fold have been reported on complex systems such as GPCRs (Joedicke et al., 2018) or ribosome nascent chain complexes (Schulte et al., 2020). This significant boost in sensitivity can be highly beneficial for real-time NMR applications. However, a sufficiently long electron relaxation for an efficient polarisation transfer to the nuclei is required, which has to be obtained under cryogenic conditions. Therefore, DNP-enhanced solid-state NMR experiments are usually performed around 100 K or even below making them less compatible with real-time studies but offering great

opportunities for cryotrapping of intermediate protein states. First attempts, initially without the help of DNP, date back to 1994 (Ramilo et al., 1994). Recently, remarkable progress has been achieved by combining rapid mixing/freeze quenching with DNP by which a time-resolution in the ms-range could be achieved (Jeon et al., 2019). Another possibility is cryotrapping of light-induced photoreceptor intermediates under DNP conditions, which is briefly addressed below.

### 3.2 Dissolution-DNP - Hyperpolarization

The application of DNP enhancement schemes employing direct microwave irradiation and use of polarizing agents is limited in liquid state NMR. Due to the large dielectric losses in water, the sample volume is usually limited to nanolitre range. However, polarization of the solvent in solid-state followed by rapid heating and injection e.g. of polarized solvent can be utilized for signal enhancement in real-time NMR measurements of biomolecules. It has been first demonstrated by Ardenkjær-Larsen (Ardenkjær-Larsen et al., 2003) and used for example to enhance the sensitivity of $^{13}$C detected *in vivo*

metabolic MRI (Kurhanewicz et al., 2019) and later developed by the Frydman group for biomolecules.

The solvent water, in a pellet form, is hyperpolarized at low temperature in a hyperpolarizer. The pellet is heated to liquid state by flushing with hot solvent. The commercially available Hypersense (Oxford Instruments Plc) instrument uses 3-4 ml of hot water while the Frydman group has developed an elegant method to reduce the dissolution factor that also extracts the radical agent, using hot non-polar solvents (Harris et al., 2011). The latter has the advantage of removing the polarizing



agent which would cause line broadening. The polarized solvent is then injected (~2 sec) into the sample already placed

inside the spectrometer.

This method has been demonstrated on the intrinsically disordered protein p27$^{Kip1}$, which refolds upon interaction with

Cdk2/cyclin (Ragavan et al., 2017). Direct polarization of p27$^{Kip1}$ in deuterated form was used to increase $T_1$ relaxation time

was.

The group of Hilty was the first to demonstrate the applicability of dissolution-DNP to folding experiments (Chen et al.,

2013). In this application, the ribosomal protein L23 was hyperpolarized and subsequently injected to folding buffer of

higher pH and folding is monitored by $^{13}$C 1D spectra.

In a more recent application, Novakovic et al. used dissolution-DNP to monitor the real-time refolding of the RNA aptamer

domain of guanine-sensing riboswitch (GSW) upon ligand binding (Novakovic et al., 2020b). Different to proteins, the

exchange of solvent water with imino sites in RNAs is sufficiently fast not only in unstructured but also in structured regions

of the RNA, and thus, all RNA sites can benefit from increase signal-to-noise due to exchange transfer. In this paper, the

GSW specific ligand (hypoxanthine) was injected parallel with the hyperpolarized solvent water to the RNA sample. Co-

injection of hyperpolarized water and refolding initiating ligand inducd refolding of the aptamer, observable with an almost

300-fold imino signal enhancement. The obtained signal enhancement is sufficient to even record series of 2D-NMR

spectrum using selective HMQC pulse sequence.

### 3.3 Photo-CIDNP

Beside microwave-driven DNP, Chemically Induced Dynamic Nuclear Polarization (CIDNP) is a method that can

selectively enhance the signal-to-noise of NMR signals. Especially photo-CIDNP offers the possibility to probe the solvent

accessibility of amino acids in proteins and peptides. During protein folding and refolding, the solvent accessible area of a

protein is changing due to hydrophobic collapse that rearranges the conformation of solvent exposed amino acids. This

probing of differential accessibility makes photo-CIDNP a powerful tool for the (real-time) investigation of protein folding

(Hore and Broadhurst, 1993; Kuhn, 2013; Morozova and Ivanov, 2019).

The positive or negative signal enhancement is based on a chemical reaction between a laser-induced excited photosensitizer

and a CIDNP-active aromatic group. The radical pair mechanism explains the photo-CINDP effect with an excited

photosensitizer that is present in the triplet state, after intersystem crossing, accepting an electron or hydrogen from a donor

molecule. A radical ion pair is formed and the singlet recombination probability of this pair is influenced by the hyperfine

coupling constants of the present magnetic field (Hore and Broadhurst, 1993). The hyperfine coupling leads to differences in

the population of the nuclear spin energy levels and therefore emissive or absorptive NMR signals, predictable by Kaptein's

rule (Kaptein, 1971). The setting of a basic photo-CIDNP pulse sequence is quite uncomplicated. After an optional

presaturation, the sample is illuminated for short time controlled by a mechanical shutter followed by the desired

experimental pulse sequence. Alternating light (with laser irradiation) and dark (without irradiation) spectra are recorded and

subtracted to obtain the difference spectra with the enhanced signals (Hore and Broadhurst, 1993). Several amino acids





exhibit polarization by photo-CIDNP in solution, these are tryptophan, tyrosine, histidine, and also methionine, glycine and methylcysteine, although to different extent (Stob and Kaptein, 1989; Morozova et al., 2005; Morozova and Yurkovskaya,

2008; Morozova et al., 2016). Embedded in a protein or peptide, Trp, Tyr, His and Met show signal enhancement if they are accessible to a photosensitizer (Kaptein et al., 1978; Hore and Broadhurst, 1993). For the photo-CIDNP reaction different dyes as photosensitizer can be used, the most common ones are substituted flavins, 2,2-dipyridyl (DP), 4-carboxy-benzophenone (4-CBP), and 3,3',4,4'-tetracarboxybenzophenone (TCBP) (Morozova and Ivanov, 2019).

The first presented photo-CIDNP studies on proteins were published by Kaptein et al. (1978) on bovine pancreatic trypsin

inhibitor (BPTI). The investigation of BPTI by photo-CIDNP showed that the enhanced signals of tyrosine residues were in line with ones exposed on the surface in the crystal structure. In the following years, the time-resolved investigation of the kinetics of folding or refolding proteins with photo-CIDNP should become more and more important. This is also due to the fact that time-resolved photo-CIDNP studies have a better time resolution than other NMR techniques, because of the laser-induced generation of nuclear polarization. The repetition rate is determined on electron relaxation, not nuclear relaxation

rates. (Day et al., 2009; Kuhn, 2013). Also the small chemical shift resolution in non-native or not folded states of proteins can be overcome due to the fact that only solvent accessible amino acids are photo-CIDNP sensitive and the investigation of unfolded or partially folded structures at a residue-specific level is possible (Schlörb et al., 2006). During folding, several conformational states including random coil, molten globule states as well as folding intermediates, non-native states, partially folded states and native states can be characterized in a residue specific manner (Kuhn, 2013).

Beside amino acids, also DNA and RNA mononucleotides are CINDP-active including guanosine, adenosine and thymidine (Kaptein et al., 1979; Pouwels et al., 1994). With a self-complementary tetramer it was shown that photo-CIDNP can only be detected in single stranded regions, when the nucleobase is accessible to the solvent and the photosensitizer (McCord et al., 1984a). photo-CIDNP-studies on tRNA were the first investigations of such kind on larger nucleic acids. Temperature-dependent photo-CIDNP experiments showed changes consistent to the melting of tertiary and secondary structures

(McCord et al., 1984b). Therefore, photo-CIDNP can also be a powerful tool for the characterization of the accessibility of nucleobases in RNA or DNA that play key roles in RNA-protein binding sites.

**4 Overview of light and rapid mixing applications**

Folding induction using rapid mixing approaches coupled to NMR is universally applicable and the installation of the rapid mixing apparatus is not costly. Experimental challenges, including deterioration of NMR spectra, remain, however. The

coupling of laser irradiation to trigger biomacromoleular folding reactions is conceptually more elegant but puts more stringent requirements to the systems under study. Advantages are obvious: the dead time of folding induction is no longer determined by built-up of NMR homogeneity and absence of flow-related susceptibility inhomogeneities across the sample, but homogeneous illumination that depends on photophysical properties, including concentration-dependent extinction coefficients. Triggering times can thus be shortened as signal-to-noise is increased. A number of biomacromolecules carry an



endogenous chromophore and selected examples of NMR spectroscopic investigations of these systems are discussed in chapter 4.1.

Other systems can be modified by photo-active non-natural groups or, alternatively cofactors or ligands can be masked by a photolabile group. These applications are discussed in chapter 4.2 along with applications of rapid mixing as the two methods complement each other on some of the systems investigated.

## 4.1 Endogenous chromophores

### 4.1.1 Photo-active yellow protein

The photo-active yellow protein (PYP) from the Ectothiorhodospira halophile shows negative phototactic response towards intense blue light (Sprenger et al., 1993). Light excitation of PYP induces a photocycle during which PYP undergoes structural and dynamic changes. The photocycle is defined by three states, the ground state pG, the red-shifted intermediate

pR and the long-lived blue-shifted intermediate pB. The photocylce is coupled to the formation of a covalently bound thiol ester-linked chromophoric group, the p-coumarin acid, which undergoes *cis-trans* isomerization upon light excitation (Hoff et al., 1994). Early on, Kaptein realized that due to the small size of the protein (14-kDa) and the chromophore, PYP is an excellent model system for the investigation of the processes occurring during photoreception in solution at atomic resolution. Kaptein's group thus investigated the blue-shifted intermediate pB of the photocycle of PYP (Düx et al., 1998;

Rubinstenn et al., 1999, 1998). By combination of light induction and NMR, the structure and backbone dynamics of the long-lived intermediate pB and the ground state pG were characterized. Irradiation with argon light inside the NMR tube generated the kinetic intermediate, while its properties were studied by multidimensional heteronuclear experiments. Besides conventional correlation and relaxation experiments, Kaptein et al. developed a 2D [1]H-[15]N-SCOTCH (Spin Coherence Transfer in Chemical Reactions) (Rubinstenn et al., 1999) experiment as a further development of the previously introduced

[1]H NMR SCOTCH experiment (Kemmink et al., 1986b). With this, the long-lived intermediate pB populated on the photocycle could be generated by light and its resonances could be correlated to the pG state as an early example of a state-correlated 2D NMR experiment. In the SCOTCH experiment, the [15]N chemical shift of pG is correlated with [1]H chemical shift of the [15]N attached proton of pB (Fig. 6 B,C). The experiments revealed that pB is structurally disordered in solution populating an ensemble of conformers in exchange on a millisecond timescale. This finding was in contrast to previous

crystal structures that showed a single structure for state pB with just minor changes around the chromophore (Genick et al., 1997). Further investigations of PYP in solution with hydrogen-deuterium exchange, pH studies, a mutant lacking negative charge and an N-terminally truncated variant revealed more detailed information about the protein and its photo active cofactor (Bernard et al., 2005; Craven et al., 2000; Derix et al., 2003).



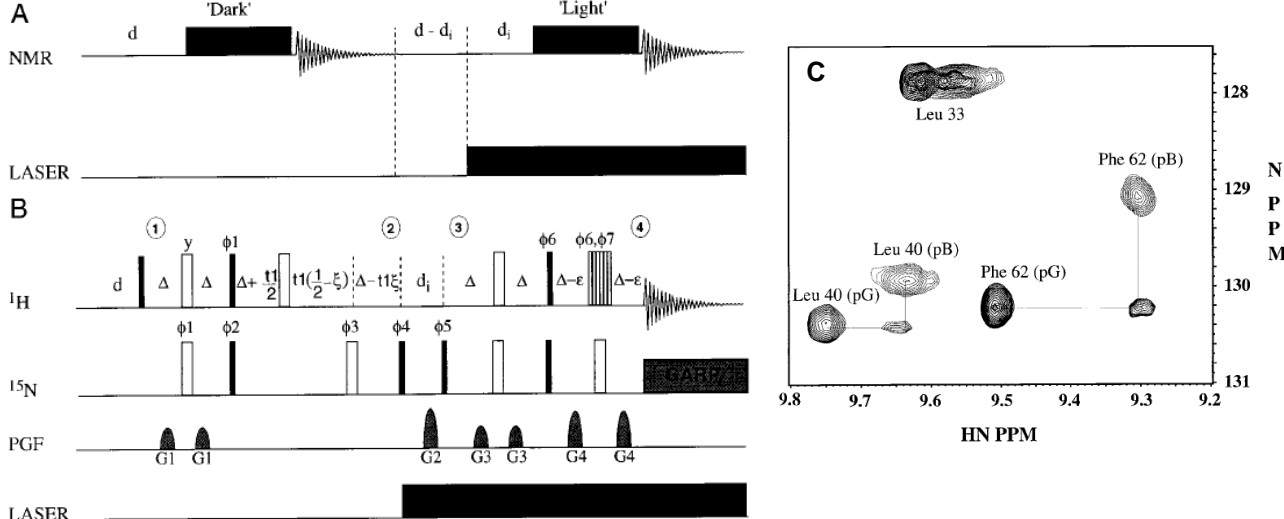

**Figure 6: General NMR scheme for the study of pB (A), $^{15}$N-SCOTCH subsequence for HSQC assignment (B). (C) Superimposed regions of the HSQC spectra of pG, pB, and the $^{15}$N-SCOTCH exchange experiment connecting the HSQC cross peaks. Reproduced with permission from Wiley (Rubinstenn et al., 1999)**

### 4.1.2 BLUF domain

Proteins containing a BLUF (sensors of **b**lue-**l**ight **u**sing **f**lavin adenine dinucleotide) domain are another representative for
proteins reacting to light. The BLUF domain is a FAD-binding domain and was found in various proteins, mainly present in proteobacteria, cyanobacteria and a few eukaryotic organism (Gomelsky and Klug, 2002). In comparison to PYP, the chromophore in BLUF domains is not covalently bound (Wu and Gardner, 2009). The proteins detect blue light using their chromophore, followed by a reversible red shift and formation of a photo-activated conformation, the signalling state that decays spontaneously to the ground state if the system returns to a dark environment (Zirak et al., 2006). Illumination
induces different structural and functional outputs as regulation of catalytic activity of enzymes and second messengers, photophobic responses and expression control of photosynthetic genes (Gomelsky and Klug, 2002). Beside other methods as time-resolved fluorescence or absorption spectroscopy, NMR coupled with light is a powerful tool for the investigation of the photoreaction mechanisms in BLUF domains after light irradiation at atomic resolution. The best characterized BLUF photoreactions analysed by NMR are the ones of AppA (Gauden et al., 2007; Grinstead et al., 2006a, 2006b), and BlrB (Jung
et al., 2005; Wu et al., 2008) (from *Rhodobacter sphaeroides*), BlrP1 (from *Klebsiella pneumonia*) (Wu and Gardner, 2009) and YcgF (from *Escherichia coli*) (Schroeder et al., 2008). Kaptein and his co-workers studied the AppA BLUF domain and presented a solution structure as well as evidence for structural changes in the light-induced state, e.g. for surface residues and the flipping of a glutamine side chains followed by the formation of a hydrogens bond (Grinstead et al., 2006b, 2006a). By mutation of aromatic amino acids that are in short distance to the FAD cofactor, they were able to gain more information
about the electron-transfer pathways in BLUF domains (Gauden et al., 2007). With NMR under light and dark conditions,



Gardner et al. observed structural changes in BlrB for amino acids near the flavin-binding pocket but also more than 1.5 nanometer apart. This finding indicates that the light-induced signal is propagated from the flavin through the protein resulting in the initiation of the regulatory function (Wu et al., 2008). Together with the Essen group in Marburg, Schwalbe

and coworkers investigated YcgF from *E. coli* (reconstituted with FMN and FAD) and in comparison to HSQC spectra of AppA and BlrB, much stronger chemical shift perturbations were observed upon light excitation (Fig. 7A) (Schroeder et al., 2008). Furthermore, they recorded kinetics for the dark state recovery of YcgF by proton NMR spectroscopy in a temperature-dependent manner. Additionally, $^{31}$P kinetic measurements were performed to investigate the kinetic behavior of the chromophore signals upon illumination (Fig. 7B-C). The results combined with UV/Vis spectroscopy show a heterogeneous distribution of half-live times for the light to dark conversion, suggesting hysteresis effects.

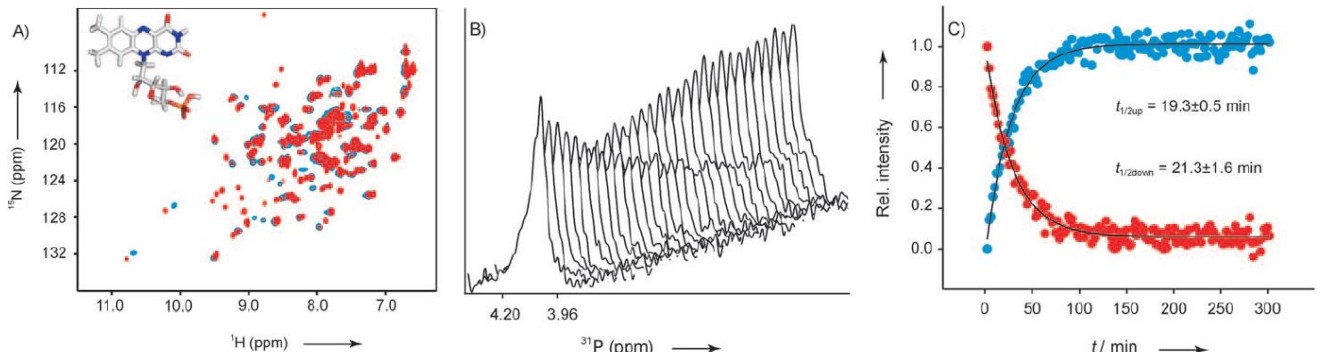


**Figure 7: a) $^1$H,$^{15}$N HSQC of YcgF (reconstituted with FMN) in the dark (blue) and light (red) state. B) $^{31}$P stack plot representations of the chromophore after light illumination. c) Normalized and fitted signal intensities of $^{31}$P kinetics traces for t1/2 calculations. Reproduced with permission from Wiley (Schroeder et al., 2008)**

**4.1.3 Time-resolved NMR studies of the photocycle of visual rhodopsins by liquid-state and solid-state NMR**

The characterization of liquid- and solid-state NMR experiments of the dark state and various light states of rhodopsins have been fascinating for long. Cryotrapping experiments on bacteriorhodopsin coupled to DNP have been pioneered in the groups of Griffin and Herzfeld (Mak-Jurkauskas et al., 2008; Ni et al., 2018). From a technical point of view, the NMR-spectroscopic time-resolved investigation of the photocycle of the eukaryotic, visual bovine rhodopsin, the mammalian visual dim-light G-protein coupled photoreceptor, represents one of the most challenging biophysical studies to characterize

key intermediates and kinetics of its photocycle as its photocycle is irreversible, highly light sensitive and the spectra of functional rhodopsin of extremely low signal-to-noise. Such functional rhodopsin can only be prepared using eukaryotic expression systems (Reeves et al., 2002).

Opsin, the apo-protein, is covalently attached to the chromophore 11-cis-retinal through Schiff base formation. Upon photon absorption, the retinal undergoes a E/Z isomerization of the cis-configured double bond to all-trans-retinal. Thus,

isomerization induces conformational transitions and population of several high-energy photocycle intermediates, whose decay leads to the formation of the meta II signal state. The aim of our time-resolved studies was the characterization of the



decay kinetics of this meta-II state resolved on individual amino acid reporter signals. For this, we could assign the $^1$H,$^{15}$N tryptophan side chain indole resonances using selectively $^{15}$N-labelled tryptophan expressed in HEK293 cells (Werner et al., 2008) and we also pursed the attachment of fluorinated reported groups to cysteine thiol groups to utilize $^{19}$F-NMR to follow

these conformational changes in a time-resolved manner (Loewen et al., 2001). After in-situ illumination of the dark state of rhodopsin, we could detect the NMR signals in the meta-II state in 2D correlation spectra and could analyse the decay kinetics of meta-II. These kinetics are bifurcated: next to the known formation of opsin, we could show the meta-III state to be populated (Fig. 8). This meta-III-state is not signalling active but considered as storage system (Stehle et al., 2014).

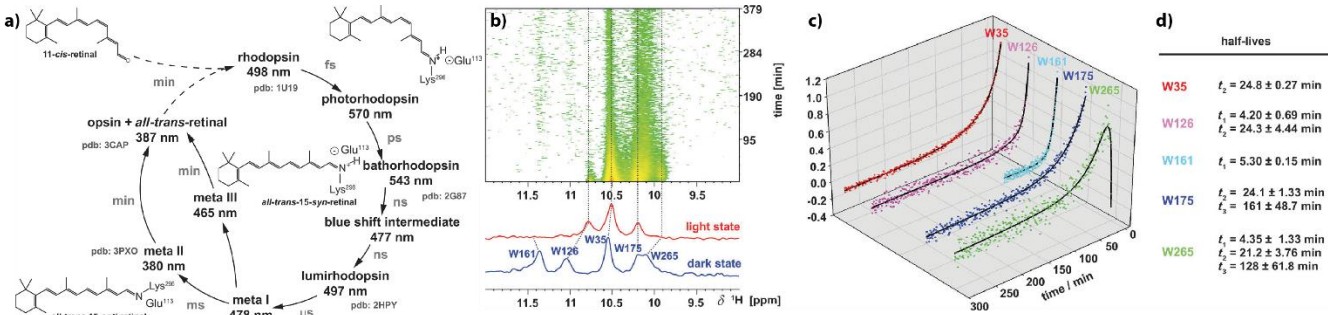

**Figure 8: Analyses of the long lived meta III and meta II states of rhodopsin after illumination. a) The photocycle of bovine rhodopsin with the first detectable intermediate of photorhodopsin and subsequent intermediates as a product of thermal relaxation. b) Time series of $^1$H-NMR spectra from the indole region of a selective α,ε-$^{15}$N-tryptophan-labeled rhodopsin and corresponding dark state (blue) and light state (red). c) Time traces of the indole signal intensities and the corresponding half-lifes extracted from exponential fits to the time dependant signal intensities. The figure was adapted with permission from Wiley**
**(Stehle et al., 2014).**

This finding is important, as continued activation of the photocycle of rhodopsin leads to the accumulation of all-trans-retinal in the rod outer segments (ROS). For retinal homeostasis, deactivation processes are required to delay the release of retinal. Bovine visual arrestin (Arr(Tr)) has been previously proposed to play a key role in the deactivation process and in fact, time-resolved NMR together with optical spectroscopy conducted by the group of Wachtveitl could show that formation of the

rhodopsin-arrestin complex markedly influences partitioning in the decay kinetics of rhodopsin. Binding of Arr(Tr) leads to an increase in the population of the meta III state that is simultaneously formed with meta II from meta I (Chatterjee et al., 2015). We further studied the retinal-disease-relevant G90D bovine rhodopsin mutant by time-resolved liquid-state and DNP-enhanced solid-state NMR with the group of Glaubitz as well as by advanced optical spectroscopy with the group of Wachtveitl (Kubatova et al., 2020). The G90D mutation is one of numerous mutations that impair the visual cycle of the

mammalian dim-light photoreceptor rhodopsin; it is a constitutively active mutant form that causes CSNB disease. Different to previous crystallographic reports, we could detect two long-lived dark states, both of which contain the retinal in 11-cis configuration. By studying the photocycle with DNP-enhanced solid-state NMR, we could detect the dark state, the bathorhodopsin and the meta-II state and could show that all these states retain their conformational heterogeneity. This conformational heterogeneity is linked to a substantially altered photocycle as shown by optical spectroscopy.



DNP-enhanced solid-state NMR in combination with cryo-trapping of light-induced intermediates of membrane-bound photoreceptors such as rhodopsins offers insight to link their 3D structures with their photochemical properties. Typical readout parameters are isotropic and anisotropic chemical shifts, homo- and heteronuclear dipole couplings or torsion angles by which finest alterations within the chromophores during the photocycle could be detected (Becker-Baldus et al., 2015; Carravetta et al., 2004; Concistrè et al., 2008). The use of DNP in these systems was demonstrated for bacteriorhodopsin

(Bajaj et al., 2009) and the discovery of many new rhodopsins inspired a series of new experiments covering the marine photoreceptor proteorhodopsin (Mehler et al., 2017), the light-gated ion channel channelrhodopsin-2[1] or the light-driven Na-pump KR2[2] (Jakdetchai et al., 2021). In order to trap a desired photointermediate for DNP solid-state NMR analysis, an optimized MAS-NMR setup is needed allowing simultaneous sample illumination by light with the desired wavelength as well as microwave irradiation (Fig. 9a). Furthermore, a suitable cryotrapping protocol has to be established, which depends

on the particular photoreceptor properties and the targeted intermediate state (see (Becker-Baldus and Glaubitz, 2018) for an overview (Mak-Jurkauskas et al., 2008; Ni et al., 2018). An illustration of this approach is provided for the light-driven proton pump proteorhodopsin (Bamann et al., 2014) in Fig. 9b-d. Proteorhodopsin is the most abundant photoreceptor found today. Light-induced cryotrapping enabled the analysis of the M-state, which is a key step for the proton transfer. This state forms after retinal isomerisation from all-*trans* to 13-*cis* and upon de-protonation of the Schiff base. The protein forms a

homo-pentamer in the membrane with functionally relevant cross-protomer interactions (Fig. 9b). A DNP-enhanced TEDOR on the trapped M-state reveals the formation of two distinct substates (Fig. 9c) (Mehler et al., 2017). The residue H75, which links the proton acceptor D97 with W34 across the protomer interface, switches its tautomeric state and ring orientation during M-state proton transfer (Fig. 9d) (Maciejko et al., 2019). Another example is shown in Fig. 9e. Channelrhodopsin-2 is a light-gated ion channel. Its photocycle is linked to channel opening and closing by a still unknown mechanism. The

chromophore retinal isomerizes upon light absorption and the protein interconverts into various photocycle intermediate states. Most of these states could be trapped under DNP conditions by thermal trapping, relaxation and freeze-quenching protocols (Becker-Baldus et al., 2015).





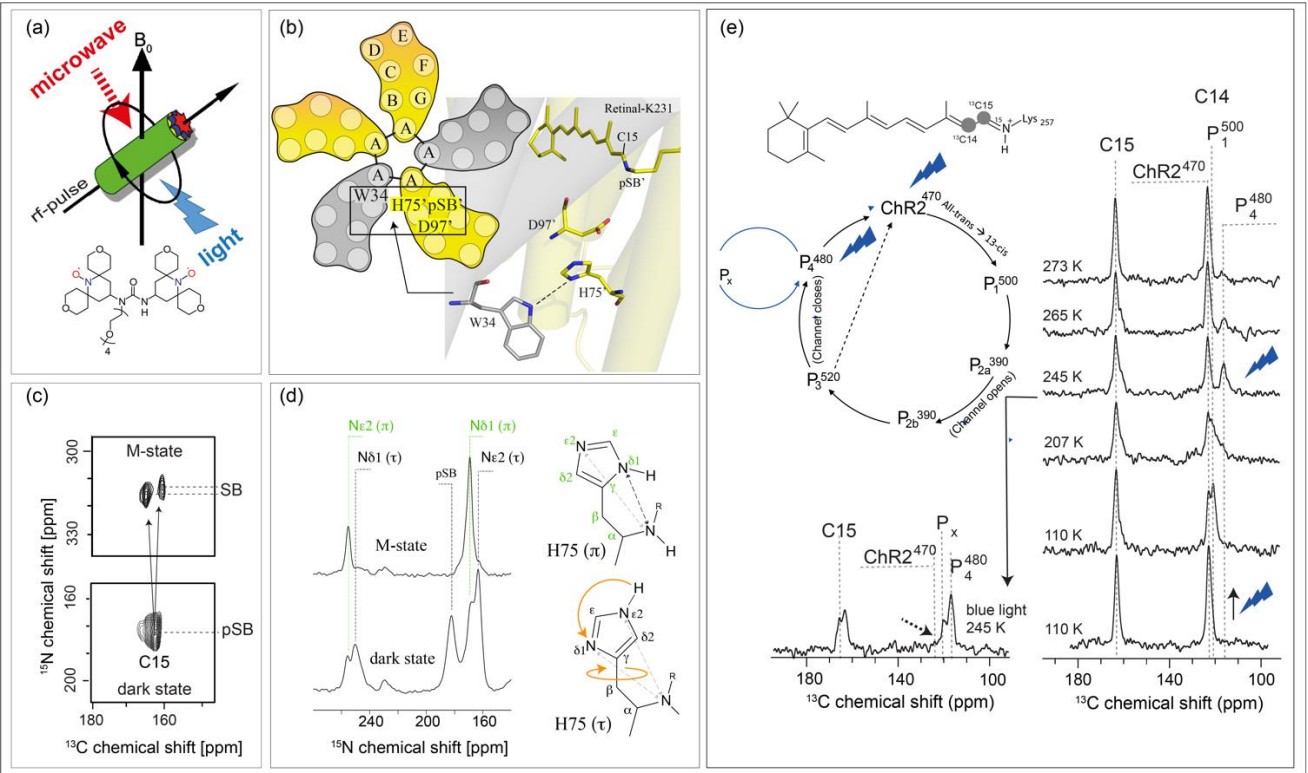

**Figure 9: Example for cryogenic trapping of light-induced photoreceptor intermediates and detection by DNP-enhanced solid-state NMR. (a) An experimental setup is required which allows simultaneous light and microwave irradiation under MAS-NMR conditions at low temperatures. (b) The pentameric proton pump proteorhodopsin undergoes a photocycle with a number of distinct intermediate states, which can be trapped for DNP solid-sate NMR (Mehler et al., 2017). (c) Upon retinal isomerisation and Schiff base deprotonation, two distinct M-states form as shown here by a NC-TEDOR spectrum. (d) In the M-state, the tautomeric and rotameric state of H75, which forms a functionally important triad with proton acceptor D97 and W34 across the pentamer interfaces, changes (Maciejko et al., 2019). (e) The application of thermal trapping, relaxation and freeze-quenching protocols together with DNP enabled the first NMR analysis of the retinal chromophore within the light-gated ion channel channelrohodopsin-2 during the photocycle (Becker-Baldus et al., 2015).**

**4.2 Rapid mixing and photochemically triggering by photolabile protecting groups**

**4.2.1 Proteins**

A number of proteins have been investigated by time-resolved NMR using mainly rapid mixing but also light induction of folding.

The small two domain calcium binding protein bovine α-lactalbumin (BLA) populates three different states depending on the buffer conditions applied: an unfolded state under denaturing conditions, a molten globule state at low pH (A-state) and a native state under native conditions. The presence of $Ca^{2+}$ stabilizes the protein and leads to increased folding rates in refolding experiments.





We investigated the Ca$^{2+}$-dependent transition from the unfolded to the folded state of BLA in the presence of urea at neutral pH using photochemical triggering by releasing Ca$^{2+}$ from the photolabile chelator DM-Nitrophen (Kühn and Schwalbe, 2000, 2000; Schlepckow et al., 2008) using laser irradiation. We could show that folding under these conditions proceeds via
parallel folding pathways (Schlepckow et al., 2008). Coupling of light-induced folding and photo-CIDNP using two lasers coupled into one quarzfiber allowed us to characterize a folding intermediate with a non-native environment that is populated already after 200 ms and has disappears again after 1.5s (Wirmer et al., 2001) see Figure 10.

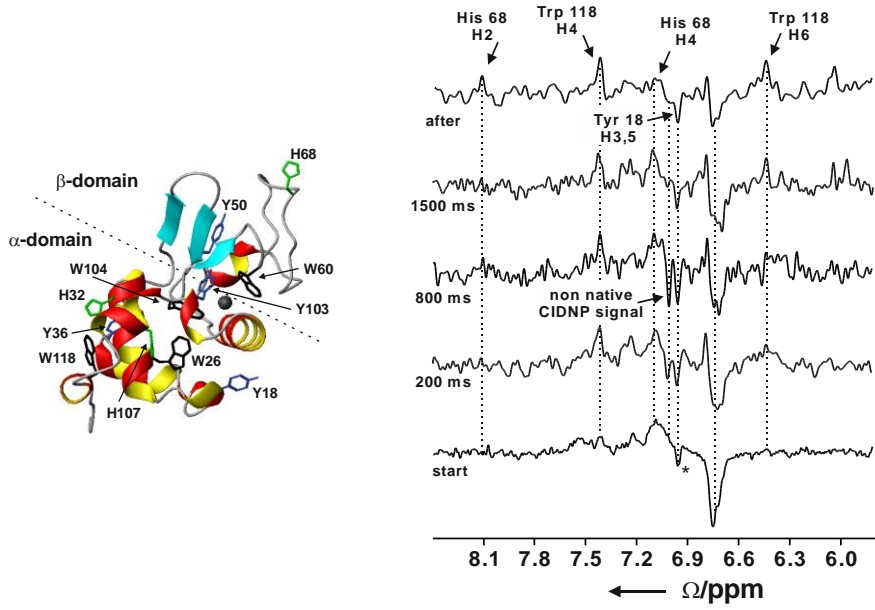

**Figure 10: Time-resolved photo-CIDNP NMR of the refolding of bovine α-lactalbumin at 4M urea upon the addition of Ca2+ using laser irradiation. Left: ribbon representation of the native structure of the protein (1HFZ.pdb) (Pike et al., 1996); * is the native signal of Tyr18 which is already present at the start of the experiment. Reproduced with permission from Wiley (Wirmer et al., 2001)**

Balbach et al. (1995) showed as early as 1995 that a folding intermediate of refolding from the GdnCl-unfolded state of α-
lactalbumin by rapid dilution resembles the molten globule of the protein by comparison of kinetic and static 1D spectra. In the second study (Balbach et al., 1996), they investigated the cooperative nature of folding from the molten globule state to the native state by raising the pH during mixing in the absence of denaturant. They extracted kinetic rates on a per residue basis from one HSQC-spectrum recorded during folding by simulating the observed line shapes. Schanda et al. directly measured folding rates of this folding process by implementing fluid turbulence-adapted SOFAST-HMQC measurements,
which allowed them to record HMQC spectra every 10.9 s during the folding process (Schanda et al., 2007). They observed uniform mono-exponential folding rates throughout the molecule confirming the presence of a single transition state.

Other proteins that were investigated using folding initiation by mixing and detection using 2D and 3D NMR spectroscopy include S54G/P55N ribonuclease T$_1$ (Haupt et al., 2011) and β$_2$-microglobulin (B2M) (Franco et al., 2017b; Rennella et al.,

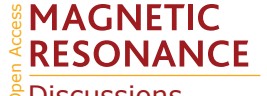

2012). B2M is an amlyloidogenic protein that folds via an intermediate which is presumably involved in the onset of
aggregation. In two impressive studies, the group of Brutscher recorded 3D BEST-TROSY experiments during the course of
folding in only 40 to 50 min and reconstructed the respective spectrum of the intermediate by comparing the real-time
spectra with static spectra. Using this methodology, they were able to obtain 3D HNCA and 3D HNCO spectra as well as
relaxation dispersion spectra of the intermediate. They could assign the intermediate and show that the intermediate is
nativelike. Furthermore, they found that the monomer-dimer transition is faster in the intermediate than in the native state.
The rapid mixing technology used for the investigation of the proteins mentioned above can be combined with photo-
CIDNP: refolding of hen egg white lysozyme (HEWL) (Dobson and Hore, 1998b; Hore et al., 1997), the histidine-
containing phosphocarrier protein HPr (Canet et al., 2003), and ribonuclease A (Day et al., 2009) have been studied.

The concept of photochemical triggering of biochemical processes by light is also appealing to solid-state NMR. So far, only
few examples have been reported in which reactions catalysed by membrane proteins have been followed by real-time MAS
NMR spectroscopy (Kaur et al., 2016; Ullrich et al., 2011). Such studies are challenging since the nature of MAS-NMR
makes the samples sealed within a MAS rotor inaccessible during the experiment preventing titration of reagents. The
reaction can therefore only be triggered for example by a T-jump on a sample mixture stored in a pre-cooled MAS rotor.
However, the feasibility to release substrates protected by photolabile groups directly during the MAS NMR experiment
followed in situ illumination has been recently demonstrated for the first time (de Mos et al., 2020): The *E. coli* lipid
regulator diacylglycerolkinase phosphorylates its lipid substrate diacylglycerol under ATP consumption. It was possible to
demonstrate that both ATP as well as the lipid substrate protected by an NPE-group can be uncaged directly under MAS-
NMR triggering DGK's enzymatic activity, which could be followed by $^{31}$P detection (Fig. 11).

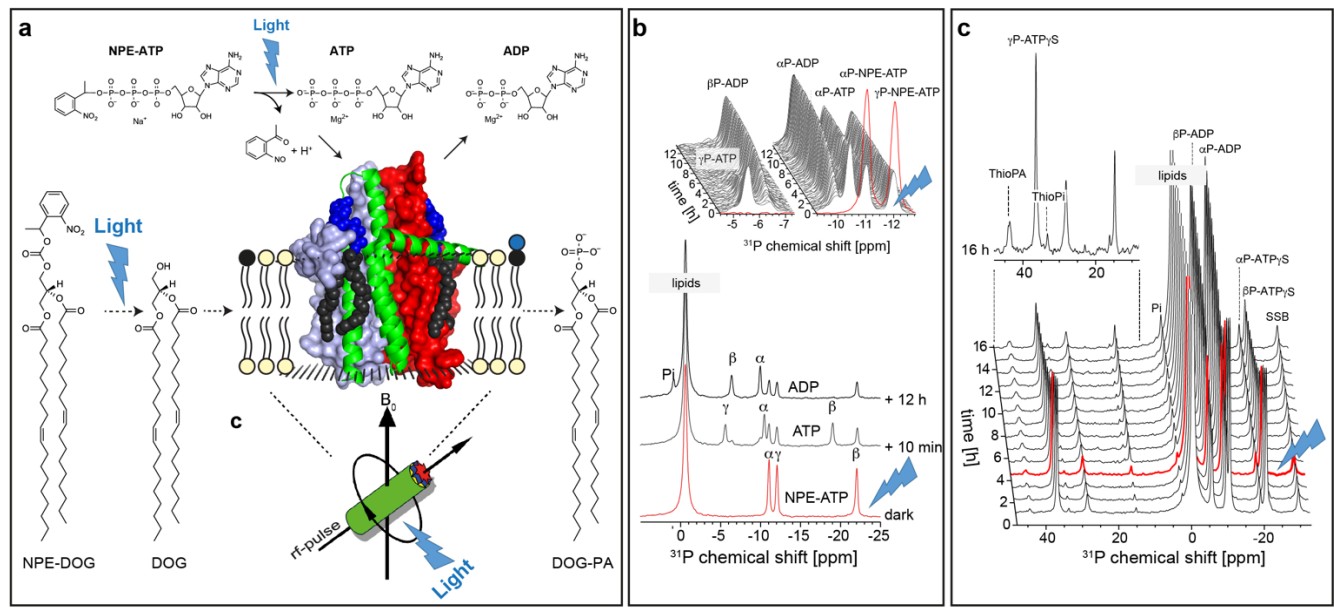





**Figure 11: Proof-of-concept demonstration for photolytic uncaging of enzymatic substrates in proteoliposomes under MAS NMR conditions (de Mos et al., 2020): (a)** *E. coli* **diacylglycerolkinase (DGK) reconstituted into liposomes containing caged lipid substrate NPE-DOG or NPE-ATP. The proteoliposomes are illuminated by UV light in situ within the MAS rotor during the NMR experiment. (b) Successful uncaging of NPE-ATP followed by ATP consumption by DGK. (c) Uncaging of NPE-DOG results in the successful formation of Thio-PA under consumption of ATP-γS.**

While the previous examples for protein folding use cofactors or internal chromophores for the initiation of folding, site specific photoprotection of amino acids has not been used in time resolved NMR, yet. A breakthrough for the site-specific labeling of proteins for NMR was the incorporation of unnatural amino acids *in vivo*. By using an orthogonal tRNA/aminoacyl-tRNA synthetase pair unnatural amino acids can be integrated in proteins in response to a TAG amber frame shift codon (Xie and Schultz, 2006). These side-specific incorporated unnatural amino acids are attractive for the

investigation of ligand binding or protein folding *in vitro* and *in vivo* (Jones et al., 2009). Especially for NMR, isotope labeled ($^{19}$F, $^{13}$C, $^{15}$N (Cellitti et al., 2008; Hammill et al., 2007; Jackson et al., 2007) photo-caged, spin-labeled and metal chelating (Lee et al., 2009; Otting, 2008; Xie et al., 2007) unnatural amino acids are of high interest. Here, we will focus on the photo-caged amino acids, represented e.g. by *o*-nritobenzyl (*o*-NB) caged tyrosine (Deiters et al., 2006), cysteine (Wu et al., 2004), lysine (Chen et al., 2009), the 4,5-dimethoxy-2-nitrobenzyl caged serine (DMNB) (Lemke et al., 2007) and 1-

Bromo-1-[4′,5′-(methylenedioxy)-2′-nitrophenyl]ethane caged selenocysteine (Welegedara et al., 2018). These caging groups have different photophysical properties: *o*-NB and the selenocysteine are cleaved by UV illumination and DMNB by blue visible light. By inserting such a mutation into a protein, the function and structure can be modified and controlled. Cellitti et al. incorporated a photo caged tyrosine into the active site of the 33 kDa thioesterase domain of human fatty acid synthase (FAS-TE) and could show inhibition of binding of the tool compound (Cellitti et al., 2008). After cleavage of the photo-cage

via UV light binding was reestablished demonstrating that site-specific labeling via photo cages can be achieved without modifying the protein sequence, but with the possibility to inhibit and regenerate the function of the natural amino acid after cleavage. Another example for inactivation of function is the use of an *o*-NB caged cysteine at the active site of a pro-apoptotic cysteine protease caspase-3. After cleavage the natural amino acid is obtained and 40% of its activity is restored (Wu et al., 2004). Photo-caged amino acids can also be used to allow for selective covalent modifications in proteins after

cleavage of the cage. With site specific incorporation of a photo-caged selenocysteine and following uncaging, it is possible to site-specific modify these due to their higher reactivity in comparison to competing cysteine residues (Welegedara et al., 2018).

**4.2.1 RNA**

In time resolved NMR studies characterising the folding or refolding of RNAs, two main strategies for the utilization of

photo-caged compounds can be employed. Either the RNA itself or a folding-inducing ligand can be modified (Fürtig et al., 2007a). Folding-inducing ligands include high affinity, low molecular-weight ligands, e.g. in riboswitch folding, or divalent ions, in particular Mg$^{2+}$. If the RNA itself is modified, several strategies can be applied with regard to choice and positioning of the photolabile functional group. One approach is to modify the nucleobases in order to sterically and chemically prevent



the formation of mutual exclusive base pairs within the different conformations whose interconversion shall be studied
(Höbartner et al., 2004). The second approach is to place the photo-cage at a functional group within the backbone of the RNA. This can be for example the 2'-OH group that is important in the establishment of stabilising interactions as well as being the mediator of RNA-catalysed reactions (Manoharan et al., 2009). For the minimal hammerhead ribozyme, caging of the active 2'OH- group at the active site in combination with position selective $^{13}$C-labelling revealed a concerted motion of both nucleotides of the catalytic centre during the catalysed cleavage reaction (Fürtig et al., 2012, 2008).

Application of the approach to cage the nucleobase led to the characterisation of refolding events in various bistable RNAs (Figure 12) (Wenter et al., 2006, 2005), to the formulation of generalized folding rules that correlate the number of base pairs to refolding rates (Fürtig et al., 2007c) and to the delineation of transition state conformations in RNA refolding reactions (Fürtig et al., 2020, 2010). When applied to RNAs for which refolding is intimately linked to biological function, the exact folding pathways during transcription involving the metastable states could be determined (Helmling et al., 2018). The
introduction of photo-caged protecting groups normally requires the production of the RNA by chemical solid phase synthesis (Brieke et al., 2012; Mayer and Heckel, 2006) rendering the simultaneous incorporation of isotope labelled nucleotides laborious and expensive (Quant et al., 1994). However, new chemo-enzymatic techniques that are able to combine chemically modified and *in vitro* transcribed, isotope labelled strands within a single RNA resolve these difficulties (Keyhani et al., 2018), and will enable light-triggered folding studies of more sizeable and complex RNAs in the future.

Likewise, tremendous advances are also made in the chemistry of photo-protecting groups utilized to cage RNAs. Whereas early studies mainly focus on the 1-(2-nitrophenyl)ethyl that needs to be cleaved with UV light and has a limited steric demand (Ellis-Davies and Kaplan, 1988), new concepts using photo-caging groups with either more red-shifted absorption or higher destabilizing potency emerge (Ruble et al., 2015; Seyfried et al., 2018). Right from the beginning, the methodology of caging interacting ligands could be utilized in the study of more complex RNA systems at the size limit of liquid state
NMR. First studies on the aptamer domain of the guanine-sensing riboswitch where the ligand hypoxanthine was caged revealed a two-state folding trajectory (Buck et al., 2007). This an important molecular feature that enables fast discrimination of cognate over near-cognate ligands and enables the kinetic control of transcription termination within the two-domain full-length riboswitch (Steinert et al., 2017). In this application, resolving the folding dynamics at the level of individual nucleotides was only possible by application of nucleotide type selective isotope labelling in conjunction with x-
filtered and X-nuclei-edited real-time NMR experiments.

Caging of divalent ions is challenging for RNA as they are often needed for proper folding of tertiary interactions but as also the affinities stay in the micro- to millimolar range. However, for the Diels−Alder ribozyme the difference in reactivity for different mutants could be traced down to the differences in local dynamics around the catalytic pocket (Manoharan et al., 2009). In this case, besides mixing also release of the divalent ions from a photo-caged chelator was possible (Fürtig et al.,
2007b).



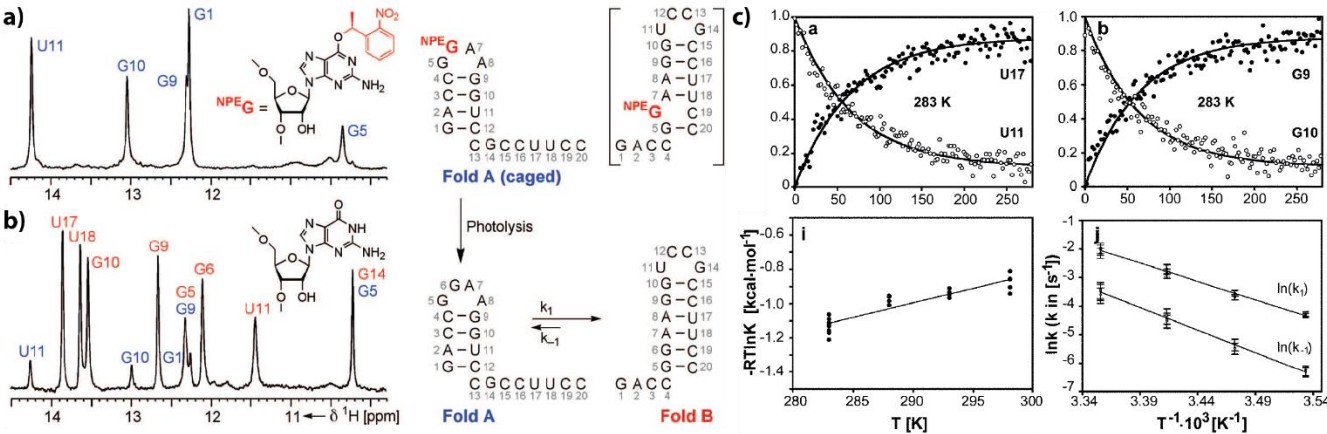

**Figure 12 a) and b) [1]H NMR spectra of imino protons with color coding according to the secondary structure shown to the right. a) Spectrum of the conformationally locked Fold A (caged by introduction of O[6]-(S)-NPE modified guanosine at position 6). c) Top: representative normalized signal intensities as a function of time after release of native state by a single laser pulse. Down: Arrhenius analyses of $k_1$ and $k_{-1}$. Reproduced with permission from Wiley (Wenter et al., 2005)**

More recently we also investigated folding of RNA using the rapid mixing methodology. Here, folding can be induced e.g. by rapid mixing of metal-ions, as has been exemplified for ribozymes by addition of $Ca^{2+}$ (Manoharan et al., 2009) and structural changes in RNA riboswitches by adding their specific ligand (Reining et al., 2013). All these applications share the detection of the kinetics on the imino-resonances in 1D spectra.

### 4.2.2 DNA and RNA non-canonical structures - Time-resolved NMR studies of DNA and RNA G-quadruplexes and DNA i-motifs folding and refolding

Non-canonical DNA structures including G-quadruplexes (G4) and i-motifs typically coexist in several heterogeneously folded conformations. This pronounced structural polymorphism and the associated inherent dynamic character make these structural motifs prime examples for time-resolved NMR studies. The pH-induced folding of a DNA i-motif revealed that the folding follows kinetic partitioning, with re-equilibration processes subsequent to the initial folding (Lannes et al., 2015; Lieblein et al., 2012). Similar findings have been made for a telomeric DNA G4 that coexists in two conformations with different folding topologies. For G4 DNA, folding can be induced by rapid injection of a $K^+$-buffer solution, since monovalent cations are essential for G4 formation. G4 folding follows complex folding pathways, which involve long-lived intermediate states that persist for several hours and the re-equilibration proceeds over days at room temperature (Bessi et al., 2015).

Recently, we investigated the folding and refolding kinetics of an 18-mer DNA (G4) forming oligonucleotide sequence from the human *cMYC* proto-oncogene promoter (Fig. 13). This G4 coexists in two conformations that are distinguished by a register shift of one G-rich strand-segment along the stacked tetrads. To study the conformational dynamics in this system, we used a combination of $K^+$-induced folding with an approach, where we photochemically trapped a single conformation



and induced refolding with *in situ* laser irradiation. By site-specific incorporation of photocages, we could block the base pair interactions for distinct nucleotides. This strategy allowed us to separate the two conformations and study the refolding mechanism in detail. The proposed kinetic model, based on kinetic and thermodynamic experimental data, reveals that after

initial folding the two conformations can directly refold into each other. The proposed transition state requires only a minimal degree of unfolding. The slow refolding kinetics (0.9 h$^{-1}$) are caused by a relatively high activation energy that is needed for an initial opening of the base paired tetrads. Further, we showed that folding kinetics induced by rapid-mixing deviate by several orders of magnitude from light induced folding. This finding highlights that the altered energy landscapes under different non-equilibrium conditions have a severe impact on the folding dynamics. Photolabile protecting groups here

are an optimal tool to investigate native, unmodified (after photocleavage) oligonucleotides dynamics under constant experimental (pressure, temperature, buffer composition) and physiological conditions (Grün et al., 2020).

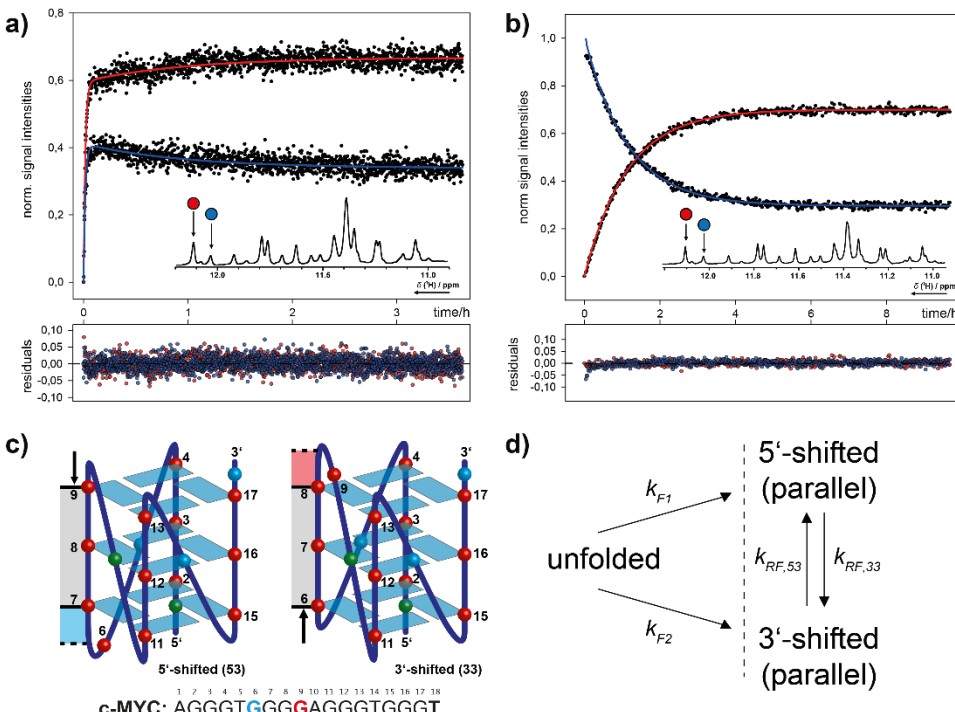

**Figure 13: Combinatorial study of folding and refolding of a 18-mer DNA G-quadruplex (G4) forming oligonucleotide sequence**
**from the human cMYC proto-oncogene promoter by (Grün et al., 2020). (a) Parallel folding of two coexistent G4 conformations, induced by *in situ* rapid-mixing with K$^+$-ions that are required for G4 formation. (b) Refolding of photocaged, isolated G4 conformations back into conformational equilibrium. (c) Schematic representation of the two coexisting G4 conformations distinguished by a register shifted G-rich strand. (d) Kinetic model for parallel folding into two coexisting conformations and subsequent refolding dynamics in conformational equilibrium. Reprinted (adapted) with permission from *J. Am. Chem. Soc.* 2020,**
**142, 1, 264–273. Copyright 2020 American Chemical Society.**

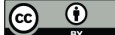

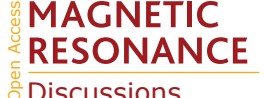

## 4.3 Overview of non-light applications

### 4.3.1 Temperature jump

There have been several different techniques for T-jump experiments in combination with real-time NMR spectroscopy, but so far applications on biomolecular systems have been limited. In most cases, RNase A was used as a test system to either
follow its heat denaturation kinetics (Akasaka et al., 1991; Naito et al., 1990) or in case of flow system (Yamasaki et al., 2013) the refolding from its heat denatured state was followed. In all cases simple two-state kinetics were observed.

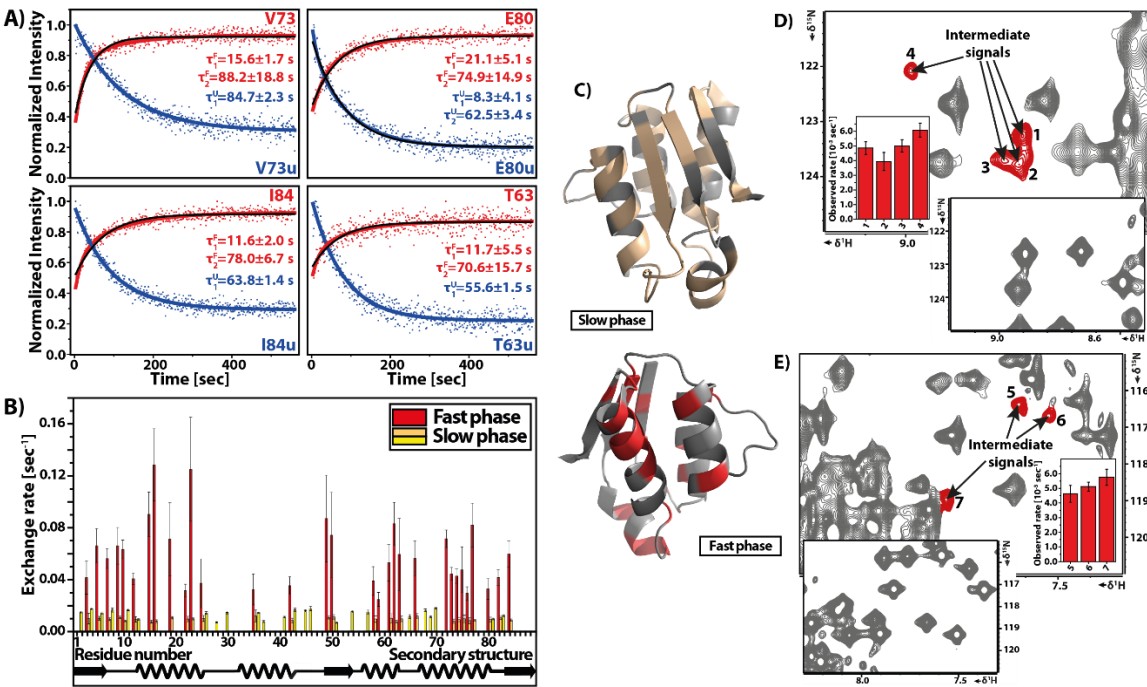

**Figure 14: T-jump experiment on the cold denatured C40A/C82A barstar mutant. A) Representative time-dependent signal intensities with double exponential kinetic fit. B) Residue specific results of refolding kinetics. C) Three-dimensional structure of
barstar (PDB: 1AB7) highlighting the affected residues with slow (pale yellow) and fast (red) phase of observed kinetics. D-E) Different part of the same $^1$H-$^{15}$N HSQC spectrum detected in slow T-jump experiment using gas-heating system with high sensitivity cryoprobe detected stable intermediate signals. Reproduced with permission from (Pintér and Schwalbe, 2020).**

The most recent application of rf-heating in combination with cold denatured barstar allowed detailed characterisation of different folding pathways. A stable intermediate was observed on the slow folding pathway (Fig. 14), where the rate
limiting step is the *trans-cis* isomerization of Tyr47-Pro48 amide bond. Additionally, the reversibility of the system and the slow *cis-trans* isomerization allowed the measurement of double-jump experiment to study alternative folding pathway. The equilibrium folded barstar was denatured by short cooling (2.5 min) time, keeping the Tyr47-Pro48 residue in *cis* conformation. From this non-equilibrium denatured state, a state-correlated spectrum was used. Although reaction rates could not be determined, evidence for an ensemble of intermediates was observed.



### 4.3.2 Pressure jump


Pressure-jump experiments have undergone rapid development in the last 10 years, thanks to the technical advancement. First Kremer et al (Kremer et al., 2011) showed in 2011 that pressure jump of around 800 bar can be achieved in both direction to de- or renature the protein. They used a model protein histidine-containing phosphocarrierprotein (HPr) to design and demonstrate new pulse sequences to study protein folding. The pressure change is an order of magnitude faster

than the longitudinal relaxation time of proteins therefore it can be directly incorporated into the pulse sequence. It is a similar approach as the SCOTCH experiment designed by Rubinstenn et al. (Rubinstenn et al., 1999) for light sensitive proteins. Their approach was either at the start of the pulse sequence (pressure perturbation transient state spectroscopy PPTSS) or during the pulse sequence (pressure perturbation state correlation spectroscopy PPSCS) to change the pressure. With these experiments they could demonstrate how pressure can be introduced as a new dimension into the pulse sequence

and measure the $k_{UN}$ and $k_{NU}$ of HPr.

Roche et al. focused on the staphylococcal nuclease (SNase) with relatively long relaxation time to the new equilibrium state (even up to 24 hours), allowing even manual pressure perturbation (Roche et al., 2013). They investigated the effect of the introduction of different cavities by mutations and their effect on the folding kinetics. They observed drastic changes both in terms of stability and folding pathways. The I92A mutant showed structurally heterogeneous ensemble at the folding barrier

with multiple folding pathways, while the WT SNase and the hyperstable D+PHS mutants have a well defined transient state and folding pathway.

The next major developments come from Charlier et al. (Alderson et al., 2017; Charlier et al., 2018a, 2018b, 2018c) in a series of publications. Their new P-jump system, as discussed in chapter 2.3 allows in both directions very fast pressure change. Their pulse sequence approaches are similar to what Kremer et al has shown already but developed even more

complex ways to incorporate into the pulse sequence and to study the folding mechanism of the pressure sensitive Ubiquitin. First the combination of H/D exchange with pressure jump revealed biexponential refolding kinetics attributed to an off pathway oligomeric intermediate. Next the use of P-jump in combination of 3D NMR spectroscopy allowed the chemical shift and $^{15}$N transverse relaxation analyses of the still unfolded protein but already at low pressure. This revealed a very short lived intermediate, which is different from the hydrogen exchange revealed one. They proposed a folding mechanism

of ubiquitin where two parallel but similarly efficient folding pathways take place: direct folding with no intermediate and folding via a short lived intermediate state. Finally, they could prove by incorporating double pressure jump into the pulse sequence and measuring the chemical shifts that this short lived intermediate closely resembles the folded state with differences as they write in "the C-terminal strand, β5, and its preceding loop, strand β1, and the C-terminal residues of strand β3, with β5 being sandwiched between β1 and β3 in the natively folded state".



### 4.3.3 "Slow" kinetics

A number of processes are slow enough to be investigated by time-resolved NMR without the need of any device to initiate folding. One example is the formation of very large macromolecular assemblies. The group of Boisbouvier studied for example the self-assembly pathway of the 0.5 MDa proteolytic machinery TET2 (Macek et al., 2017), the group of Schwarzer investigated histone modification by time-resolved NMR-spectroscopy (Liokatis et al., 2016).

The investigation of protein modification and in particular phosphorylation has been put forward by Selenko in 2010 and been used since than in a couple of applications (Kosten et al., 2014, p. 129; Landrieu et al., 2006; Liokatis et al., 2010; Mylona et al., 2016). Such studies are even possible in cellular environment by time-resolved in-cell NMR (Theillet et al., 2013). Other applications of time-resolved in-cell NMR are the proteolytic alpha-synuclein processing (Limatola et al., 2018), the methylation of lysines in cell, the investigation of ligand binding in cellular environment (Luchinat et al., 2020a, 2020b) and the modulation of bound GTP levels of RAS (Zhao et al., 2020). Time-resolved NMR experiments are now also pursued to characterize metabolic flux in patient-derived primary cells (Alshamleh et al., 2020; Reed et al., 2019).

The formation of amyloids is another highly relevant slow process. While fast tumbling monomers and flexible tails are amenable to liquid state NMR spectroscopy, residues in intermediate exchange present in oligomers or residues in fibrils cannot be observed. Therefore, the aggregation is monitored as monomer loss kinetics following the signal decrease of the resonances present at the beginning of the experiment.

The misfolding of the prion protein into fibrils is observed in neurogenerative prion diseases. While the prion protein is a mainly α-helical globular protein with an unfolded tail in its native state, β-sheet structures are enriched in the polymeric fibrillar forms. We investigated the kinetics of fibril formation from the unfolded state on a per residue basis of human prion protein (Kumar et al., 2010) and murine prion protein (Schlepckow and Schwalbe, 2013). Comparison of HSQC spectra directly dissolving the human protein (90-230) and after 4 and 7 days reveals that signals are lost fast for the core of the fibril (145-223), while N-terminal signals decreases slower and signals close to the C-terminal (224-230) change their chemical shift indicating a structural change in the latter region within the fibril. A more detailed view of the fibril formation was obtained on the murine prion protein where signal loss rates could be obtained from multiple HSQC measurements on a per residue basis. Here we found that residues in close proximity to the disulfide bridge (C179-C214) broaden first which we attribute to initial molecular contacts in oligomer formation, while in a second stage of the aggregation fibrils are formed.

Another disease attributed to protein misfolding is the Alzheimer's disease, in which Aβ peptides are forming fibrils. The aggregation of Aβ1-40 and Aβ1-42 has been investigated by a number of groups (Bellomo et al., 2018; Pauwels et al., 2012; Roche et al., 2016) by monitoring the loss of the monomers. Using the decay of the methylgroup region in a proton 1D as reporter of aggregation, Luchinat and coworkers (Bellomo et al., 2018) conducted extensive parallel simulations on the folding kinetics at different initial monomer concentrations applying a range of different models. Kinetics of amyloid formation can be best described by a model in which oligomers are formed which transform irreversibly into fibrils at a



certain oligomer size. These fibrils grow (by addition and release of monomers) and undergo fibril fragmentation resulting in smaller fibrils that in turn grow further.

Switching gears completely: another impressive example for the application of liquid NMR spectroscopy for the investigation of "slow" kinetics is the study of tRNA maturation. Barraud et al. (2019) were able to investigate the enzymatic modification of tRNA$^{Phe}$ in yeast cell-extract over 26 h using a series of HSQC spectra. Figure 15 shows HSQC spectra of the maturation after 12-14 h after addition of the tRNA$^{Phe}$ to yeast extract showing that modifications are inserted in a specific order along a defined route. This application nicely demonstrates the power of NMR to investigate complex mechanisms on the per site resolution.

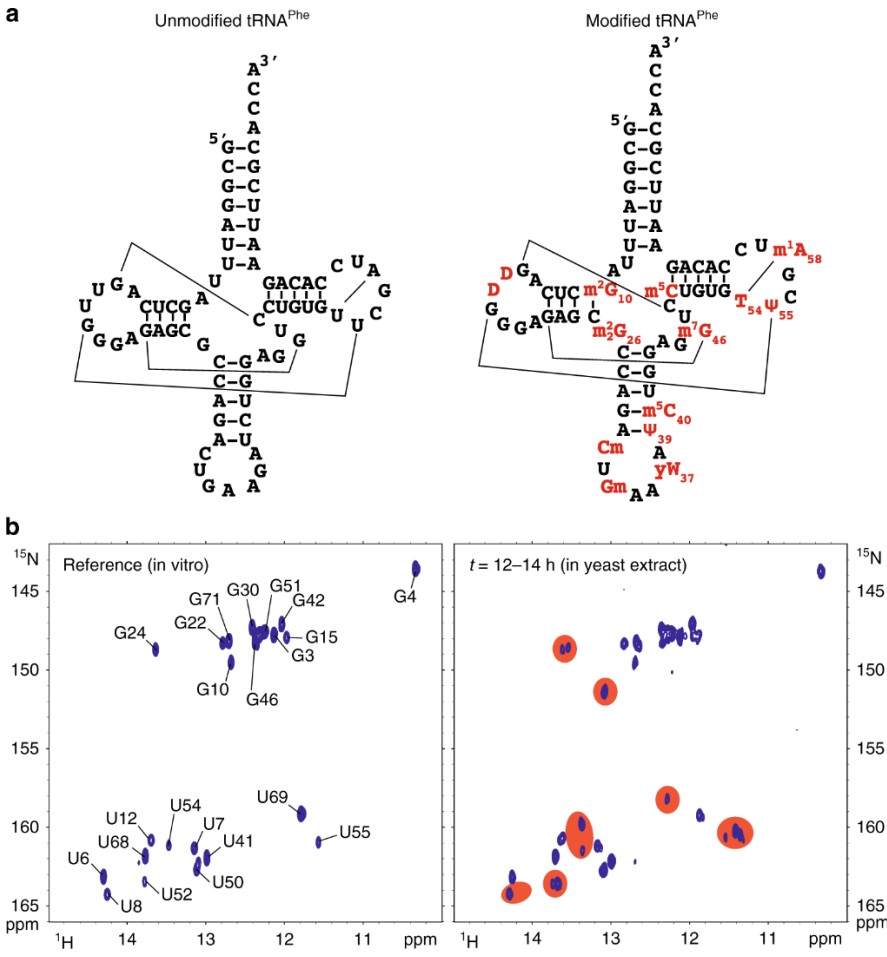

**Figure 15: A RNA secondary structure of unmodified and modified tRNAP$^{he}$a sequence and cloverleaf representation of unmodified yeast tRNA$^{Phe}$ (left) and modified tRNA$^{Phe}$ (right). b) $^1$H-$^{15}$N HSQC spectra in vitro (left) and upon 12 h after addition to yeast extract. Assignments are indicated in the in vitro spectrum while red patches highlight areas with significant changes. Reprinted from (Barraud et al., 2019)**



# 5 Conclusions

In this review, we have discussed the application of time-resolved NMR studies to study biomacromolecular folding, refolding, modification and aggregation. These studies utilize the power of NMR spectroscopy to determine kinetics of structural transitions together with site-resolution. Different to other structural techniques, NMR does not only provide snapshots of folding trajectories, but provides positive evidence for the transition of two or several conformational states. It can determine the associated kinetic rates with rates as fast as 5000 $s^{-1}$ in a significant temperature range, allowing to classify structural transitions to follow Arrhenius or non-Arrhenius behaviour. Particularly interesting are biomolecular systems whose folding trajectory is subject to kinetic partitioning. Starting from a single state, folding pathways diverge, multiple folding pathways are populated, with kinetically or thermodynamically driven conformational states. Together with mutational studies, NMR is key in delineating transitions state characteristics and to detected lowly populated states and prime examples have been reported for proteins (Korzhnev et al., 2004) and for DNA (Kimsey et al., 2018) and their complexes (Afek et al., 2020). Future applications, thanks to unstoppable developments to increase signal-to-noise and resolution in NMR, will devise more sophisticated experiments to characterize transient conformations that often represent the key states carrying the biomolecular function.

# 6 Acknowledgements

We wish to acknowledge numerous collaborations with scientists that contributed ideas to the work present. In particular, we wish to express our gratitude to R. Kaptein and R. Boelens for their help in early days of time-resolved NMR spectroscopy in the groups of the authors. Funding is acknowledged from Deutsche Forschungsgemeinschaft (CRC902, GRK1986, Normalverfahren), European commission (ITN, iNEXT, iNEXT-discovery), German Cancer research center (DKTK), and the state of Hesse (BMRZ).

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
