# Peer review of "Real-time NMR spectroscopy in the study of biomolecular kinetics and dynamics"

_Magnetic Resonance, 2021_

## Author Response (AR1)

**FIRST REVIEWER (P.J. HORE) COMMENTS AND ANSWERS:**

Lines 50, 53, 55: by describing photo-CIDNP dyes as "fluorescent" and "fluorophores" you give the impression that fluorescence if somehow essential for the generation of nuclear polarization. Efficient fluorescence in competition with intersystem crossing would, of course, be a distinct disadvantage in sensitivity terms.

We thank the reviewer for pointing out, we rephrased this part to avoid confusion.

Line 54: did you mean to include phenylalanine amongst the list of polarizable amino acids?

No, we deleted it.

Line 183: photoactive yellow protein?

Line 187: shouldn't free enthalpy be simply enthalpy or enthalpy change?

Line 278: dilution factor?

Lines 283-284: incomplete sentence?

Lines 304 and 330: CIDNP not CINDP

Line 361: irradiation with argon-ion laser light?

Line 456: channel rhodopsin

Line 548: nitrobenzyl

Lines 306-307: "influenced by the hyperfine coupling constants of the present magnetic field" makes no sense.

We thank the reviewer for pointing out these formatting and spelling mistakes, we corrected the corresponding words and sentences.

Line 670: longitudinal proton relaxation time?

We added "longitudinal $^{15}N$ relaxation", as these experiments are usually utilizing the advantageous $^{15}N_z$ spin state to store magnetization during the trigger time and/or during additional longer waiting periods.

Not all abbreviations are defined (or used more than once). While I think this is acceptable for NMR pulse sequence acronyms, there are others which will be less familiar to readers of this review, for example: FAD, FMN, GPCR, TFE, DMSO, CSNB, D+PHS, RAS, TET2, ...

We wrote out further abbreviations where the meaning is unclear. For some of the protein names, we find it unfeasible but at least the protein family where they belong is stated in the text.

**SECOND REVIEWER (K. AKASAKA) COMMENTS AND ANSWERS:**

1. **General comments:**

As the title shows, this review article covers the recent developments of NMR methodology and its application in the field of bio-macromolecular (proteins, RNA and DNA) dynamics related to both reversible and irreversible folding/unfolding phenomena and basically reversible conformational changes in bio-macromolecules.

This article is unique in focusing on a wide variety of dynamic NMR methodologies developed in last ~30 years or so for the purpose of studying biomolecular dynamics of protein, RNA and DNA for understanding their folding, unfolding, conformational dynamics, kinetics and aggregation, preferably in atomic resolution detail.

In section 2, apart from discussions on thermodynamic and kinetic information to be gained, the review goes directly into novel experimental methodologies with hardware development and pulse sequences. Particularly, the methodologies for inducing non-equilibrium conditions in target macromolecular systems and the associated rf pulse sequences for NMR observation. Representative hardware setups for triggering non-equilibrium conditions are shown by photos, and the strategic rf pulse sequences for observing time-dependent NMR signals are given in another figure (Figure 5 along with Table 1). For time-resolved irreversible experiments, which do not allow signal accumulations, a few alternative methods for increasing effective NMR sensitivity are introduced in section 3. In section 4, various time-resolved NMR spectroscopies are used to follow various protein dynamics including

light and rapid mixing applications. photo-CIDNP for studying dynamics in proteins. The method has been extended to include nucleic acids.

In summary, the review is an excellent contribution regarding the major topics and advancement in recent years pertaining to the title issue, giving an overall view of the frontier field of NMR spectroscopy and biopolymer science.

Unfortunately, there appears some lack of recognition by the authors on the historical background in some area of science, which I would like to put some comments on, in addition to several minor errors which I'd like to be corrected.

1. **Questions and suggestions**

Question 1.

Are the folding intermediates detectable only by the slow time-dependent experiment, i.e., by real-time NMR ?

The authors' comment below (in Page 3, lines 86-89) seems to support this:

"While equilibrium studies focus on characterization of conformational transitions in the microsecond-to-millisecond time scale involving NOESY-type experiments (Evans et al., 1989), line shape analysis (Evans et al., 1989; Huang and Oas, 1995) or relaxation dispersion (Korzhnev et al., 2004), non-equilibrium studies focus on slower biomolecular folding transitions."

(By this comment, the authors seem to be telling that the information on the folding intermediates may be obtained only from slow time-dependent experiments, namely, the real-time NMR spectroscopy.)

This statement does not apply to the high-pressure NMR experiment of proteins performed under static pressures. Indeed, we have practically overcome this limitation already in 2003, by utilizing the high-pressure NMR spectroscopy carried out under elevated, but static pressures; Indeed, we found;

1) At certain levels of pressure, we detect signals from "intermediate" conformers.

2) In most cases, the equilibrium "intermediates" are expected to be closely identical in structure with the kinetic intermediates detected in kinetic folding experiments (cf. for example, see Kitahara, R., Akasaka, K. (2003), Proc. Natl. Acad. Sci. 100, 3167–3172; Kitahara et al. (2005). NMR Snapshots of a Fluctuating Protein Structure: Ubiquitin at 30 bar–3 kbar. J. Mol. Biol. 347, 277–285).

In general, with the high-pressure NMR spectroscopy carried out under static hydrostatic pressures, we can identify intermediates as distinctly different thermodynamic entity from that of the basic folded conformer, and determine their structures stably trapped under pressure in fair detail  (cf. Akasaka, K., Kitahara, R., Kamatari, Y. O, (2003), in Advances in High Pressure Bioscience and Biotechnology II (R. Winter, Ed.) pp. 9-14; Akasaka, K. (2018), Protein Studies by High Pressure NMR. in Experimental Approaches of NMR Spectroscopy (Chapter 1), pp. 1-33, Springer). Under the circumstance, combining the high-level structural information along with thermodynamic information obtainable from the "static" high-pressure NMR experiment (Akasaka, K. (2003). Pure Appl. Chem., 75, 927–936.; Akasaka, K. (2003). Biochemistry, 42(37), 10875–10885); Akasaka, K. (2006). Chemical Reviews, 106, 1814–1835) with the information obtainable from the kinetic information from real-time NMR spectroscopy as exemplified in this review seems to me to a promising way toward better understanding of protein folding studies.

Indeed as the reviewer mentions intermediate states can be stabilized and detected by changing the pressure conditions. Therefore, we added the following citations as examples to the manuscript in line 216:

- Kitahara, R., Akasaka, K. (2003), Proc. Natl. Acad. Sci. 100, 3167–3172
- Kitahara et al. (2005). NMR Snapshots of a Fluctuating Protein Structure: Ubiquitin at 30 bar–3 kbar. J. Mol. Biol. 347, 277–285)

And in line 219:

- Akasaka, K. Probing Conformational Fluctuation of Proteins by Pressure Perturbation. Chem. Rev. 2006, 106 (5), 1814–1835.
- Lassalle, M. W.; Akasaka, K. The Use of High-Pressure Nuclear Magnetic Resonance to Study Protein Folding. In Protein Folding Protocols; Bai, Y., Nussinov, R., Eds.; Methods in Molecular BiologyTM; Humana Press: Totowa, NJ, 2006; pp 21–38.
- Akasaka, K. Protein Studies by High-Pressure NMR. In Experimental Approaches of NMR Spectroscopy: Methodology and Application to Life Science and Materials Science; The Nuclear Magnetic Resonance Society of Japan, Ed.; Springer: Singapore, 2018; pp 3–36.

Question 2.

Proper reference citation requested for the "State-Correlated 2D NMR Spectroscopy"

In Page 11 (both in the text and in Figure legend (Fig. 3)), the term "State-Correlated (SC)" spectroscopy is mentioned, Here, no reference is made to, the original paper in which the first "State-Correlated (SC)" spectroscopy was carried out with its naming (the paper (Naito, Nakatani, Imanari and Akasaka, J. Magn. Reson. 1990, 87 429-432)).

Early history of State-Correlated 2D NMR Spectroscopy:

In 1989, our idea was to demonstrate a new type of two-dimensional NMR spectroscopy using our newly developed microwave T-jump NMR apparatus, which would correlate the states of spin magnetization of a molecule embedded in two different thermodynamic states. In 1990, the first such experiment was performed to correlate the signals of water protons at two different thermodynamic states (in fact, two different temperatures), which was performed with the home-made microwave T jump apparatus with an innovative design employing the microwave heating assisted by a dielectric resonator (cf. Kawakami, M., & Akasaka, K. (1998). Microwave temperature-jump nuclear magnetic resonance system for aqueous solutions. Review of Scientific Instruments, 69(9). https://doi.org/10.1063/1.1149102), We were successful in obtaining the first two-dimensional spectrum of water protons connecting the two temperatures and published the result with the title "State-Correlated (SC) two-dimensional NMR spectroscopy" (Naito, Nakatani, Imanari and Akasaka, J. Magn. Reson. 1990, 87 429-432). One year later, in 1991, the first SC-2D experiment was performed on a protein in solution (cf. Akasaka, K. et al., Novel method for NMR spectral correlation between the native state conformer and the heat-excited state (probably closely heat denatured state) of protein ribonuclease A. J. Am. Chem. Soc. 1991, 113, 4688). Later, the method was extended to separate signals in liquid crystals (cf. Naito et al., J. Chem. Phys.1996,105, 4504).

Proper reference citation requested:

- I would appreciate the authors to cite this original and classical article on the SC-2D spectroscopy, Naito, Nakatani, Imanari and Akasaka, J. Magn. Reson. 1990, 87 429-432, in the text (line 243) and in the legend of Figure 5 in page 11.

We are thankful for the review for his comments, and we corrected the manuscript by adding the citations in line 319 as well as in Figure 5 caption, line 505.

- Is it possible to revise the figure in Fig. 5 for SC-2D experiment to be remade in the direction to cope with the standard 2D NMR spectroscopy (by Richard Ernst) like that in Fig. 1, Akasaka et al. (1991). J. Am. Chem. Soc. 113, 4688-4689?

We thank the reviewer for his suggestion and considering the advancement of such correlation experiments between two physically different equilibria in combination with different types of triggers we think our figure captures the general idea better but does not exclude other literature known pulse sequences (listed below) based on state-correlated spectroscopy.

- Rubinstenn, G.; Vuister, G. W.; Zwanenburg, N.; Hellingwerf, K. J.; Boelens, R.; Kaptein, R. NMR Experiments for the Study of Photointermediates: Application to the Photoactive Yellow Protein. Journal of Magnetic Resonance 1999, 137 (2), 443–447

- Charlier, C.; Alderson, T. R.; Courtney, J. M.; Ying, J.; Anfinrud, P.; Bax, A. Study of Protein Folding under Native Conditions by Rapidly Switching the Hydrostatic Pressure inside an NMR Sample Cell. PNAS 2018, 115 (18), E4169–E4178.
- Charlier, C.; Courtney, J. M.; Alderson, T. R.; Anfinrud, P.; Bax, A. Monitoring 15N Chemical Shifts During Protein Folding by Pressure-Jump NMR. J. Am. Chem. Soc. 2018, 140 (26), 8096–8099.
- Pintér, G.; Schwalbe, H. Refolding of Cold-Denatured Barstar Induced by Radio-Frequency Heating: A New Method to Study Protein Folding by Real-Time NMR Spectroscopy. Angewandte Chemie International Edition 2020, 59 (49), 22086–22091.

- **Corrections/Additions**

Page 4; lines 100-105; The general statement here on the effect of pressure-jump and temperature-jump on biomolecular folding and refolding appears inappropriate. The effect of pressure on protein conformational stability is primarily determined by the volume change of the system, while the effect of temperature on the conformational stability is primarily determined through the heat capacity change of hydration.

In line 100-105 temperature and pressure change is only introduced as a mean to change conformation as suitable trigger for RT-NMR investigation. We would like to keep the review easy to read for the wider audience therefore we would avoid in depth thermodynamic discussions.

Page 5: lines 137-145; The effect of pressure on bio-macromolecular structure in aqueous solution should better be discussed on thermodynamic terms. The literatures cited here are not considered appropriate. I urge the authors to reconsider the citations here to include the more fundamental ones.

Additional citations for more detailed descriptions about high pressure NMR and its theoretical/thermodynamic background is now cited in line 214.

Akasaka, K. Probing Conformational Fluctuation of Proteins by Pressure Perturbation. *Chem. Rev.* **2006**, *106* (5), 1814–1835.

Lassalle, M. W.; Akasaka, K. The Use of High-Pressure Nuclear Magnetic Resonance to Study Protein Folding. In *Protein Folding Protocols*; Bai, Y., Nussinov, R., Eds.; Methods in Molecular Biology™; Humana Press: Totowa, NJ, 2006; pp 21–38.

Akasaka, K. Protein Studies by High-Pressure NMR. In *Experimental Approaches of NMR Spectroscopy: Methodology and Application to Life Science and Materials Science*; The Nuclear Magnetic Resonance Society of Japan, Ed.; Springer: Singapore, 2018; pp 3–36.

Page 11; Figure 5.

All figures in Fig. 5 are difficult to follow both in size and logic. They should be enlarged and the pulse sequences with comments should be clearer with certain explanation.

Thank you for the reviewer comments, we have adapted the figure to be more clear for the reader.

Page 27; line 650; (T-jump)

Please correct the following citation:

Naito et al.(1990) should be replaced by Kawakami, M., Akasaka, K. Review of Scientific Instruments 69, 3365 (1998)

==We thank the reviewer for finding this mistake, we have corrected the citation.==

Page 29; 4.3.3 "Slow" kinetics (P-jump)

For your information, some additional references (1-3) related to the topics "Slow" kinetics (P-jump) are given below. They are all on "Slow" pressure-jump NMR studies of amyloid fibrils, showing observation of pressure-induced dissociation or re-association of amyloid fibrils (1, 2) and prion fibrils (3) with real-time NMR spectroscopy: If you consider worth citation of any of them, please go ahead.

1) Kamatari, Y. O., Yokoyama, S., Tachibana, H., & Akasaka, K. (2005). Pressure-jump NMR Study of Dissociation and Association of Amyloid Protofibrils. In Journal of Molecular Biology (Vol. 349, Issue 5).

2) Niraula, T. N., Konno, T., Li, H., Yamada, H., Akasaka, K., & Tachibana, H. (2004). Pressure-dissociable reversible assembly of intrinsically denatured lysozyme is a precursor for amyloid fibrils. Proceedings of the National Academy of Sciences of the United States of America, 101(12), 4089–4093.

3) Kazuyuki Akasaka, Akihiro Maeno, Taichi Murayama, Hideki Tachibana, Yuzo Fujita, Hitoki Yamanaka Noriyuki Nishida & Ryuichiro Atarashi (2014). Pressure-assisted dissociation and degradation of "proteinase K-resistant" fibrils prepared by seeding with scrapie-infected hamster prion protein, Prion, 8:4, 314-318.

==We have added and shortly discussed the suggested citations in chapter 4.3.2. in line 676-679.==

Finally, it is a great pleasure for me to be given this opportunity of commenting on this review article prepared by the outstandingly active research group in Frankfurt guided by Harald Schwalbe. In particular, on this special occasion celebrating

Robert Kaptein, a rare leader in biological NMR spectroscopy, for his 80[th] birthday

Kazuyuki Akasaka, Ph.D.